# High-resolution simulations of interactions between surface ocean dynamics and frazil ice

Agnieszka Herman[1], Maciej Dojczman[2], and Kamila Świszcz[1]

[1]Institute of Oceanography, University of Gdansk, Poland
[2]Faculty of Physics, University of Warsaw, Poland

**Correspondence:** Agnieszka Herman (oceagah@ug.edu.pl)

**Abstract.** Frazil and grease ice forms in the ocean mixed layer (OML) during highly turbulent conditions (strong wind, large waves) accompanied by intense heat loss to the atmosphere. Three main velocity scales that shape the complex, three-dimensional OML dynamics under those conditions are: the friction velocity $u_*$ at the ocean–atmosphere interface, the vertical velocity $w_*$ associated with convective motion, and the vertical velocity $w_{*,L}$ associated with Langmuir turbulence. The fate of buoyant particles, e.g. frazil crystals, in that dynamic environment depends primarily on their floatability, i.e., the ratio of their rising velocity $w_t$ to the characteristic vertical velocity, dependent on $w_*$ and $w_{*,L}$. In this work, dynamics of frazil ice is investigated numerically with a high-resolution, non-hydrostatic hydrodynamic model CROCO (Coastal and Regional Ocean COmmunity Model), extended to account for frazil transport and its interactions with surrounding water. An idealized model setup is used (a square computational domain with periodic lateral boundaries; spatially uniform atmospheric and wave forcing). The model reproduces the main features of buoyancy- and wave-forced OML circulation, including the preferential concentration of frazil particles in elongated patches at the sea surface. Two spatial patterns are identified in the distribution of frazil volume fraction at the surface, one related to individual surface convergence zones, very narrow and oriented approximately parallel to the wind/wave direction, and one in the form of wide streaks with separation distance of a few hundreds meters, oriented obliquely to the direction of the forcing. Several series of simulations are performed, differing in terms of the level of coupling between the frazil and hydrodynamic processes: from a situation when frazil has no influence on hydrodynamics (as in most models of material transport in the OML) to a situation when frazil modifies the net density, effective viscosity, transfer coefficients at the ocean–atmosphere interface and exerts a net drag force on the surrounding water. The role of each of those effects in shaping the bulk OML characteristics and frazil transport is assessed, and the density of the ice–water mixture is found to have the strongest influence on those characteristics.

## 1 Introduction

Seasonal sea ice cover in polar and subpolar seas is an important mediator of heat, moisture and momentum exchange between the atmosphere and the ocean, and thus an indicator of short- and long-term changes of weather and climate. [c1] Physical processes involved in shaping the evolution of sea ice are intrinsically coupled to accompanying processes in the lower atmosphere

---

[c1] ~~Obviously,~~

and the upper ocean – sea ice forms and evolves at the interface of two turbulent boundary layers, understanding and modeling of which is a challenge in itself (e.g., McPhee, 2008)[c2]. The nature of the ocean–sea ice–atmosphere coupling changes with the seasons and is very different under freezing conditions, during periods of new ice formation, and under melting conditions, when a relatively thick ice cover undergoes fragmentation and melting. The conditions of interest in this paper are those that

accompany sea ice formation under turbulent conditions, especially in latent heat polynyas and similar, spatially limited water bodies, forced by strong winds and large heat loss to the atmosphere. A distinctive feature of those regions is turbulent mixing related to wind shear, locally generated wind waves and thermal convection (Morales Maqueda et al., 2004)[c3], which together tend to produce a deep and cold ocean mixed layer (OML) and lead to formation of small ice crystals, so-called frazil ice, within the water column (as opposed to freezing at the surface).

Understanding physical processes accompanying frazil ice formation in turbulent OMLs is important for a number of reasons. Reliable parameterizations of those processes are necessary for continuum numerical sea ice models to provide realistic estimates of ice cover growth and OML evolution in freezing periods. In state-of-the-art sea ice models, rather crude algorithms based on basic thermodynamic principles are used to "produce" new ice, which is then assumed to form a thin, uniform layer on the surface (grease ice), with either constant or variable thickness (Biggs and Willmott, 2000; Vancoppenolle et al., 2009).

In spite of recent achievements in developing frazil and grease ice parameterizations for leads and polynyas (Wilchinsky et al., 2015) and in numerical modelling of frazil formation in general (e.g., Heorton et al., 2017; Rees Jones and Wells, 2018, see also references there), several important aspects of those processes remain unexplored and therefore not included in [c1]continuum sea ice models. Observations in polynyas and in the marginal ice zone (MIZ) clearly show that in typical conditions, under the action of wind, waves and atmospheric cooling – we will see later that the combination of those three factors is essential

for determining the OML mixing "regime" – a complex, three-dimensional circulation develops within the OML, affecting the formation and spatiotemporal distribution of frazil crystals. In turn, the presence of frazil [c2]influences the details of that circulation, for example by modifying water density, effective viscosity or air–ocean heat and momentum transfer. The exact mechanisms underlying those mutual interactions and, crucially, their importance at larger scales remains to be investigated. There are two reasons to assume that significant progress will take place in that research area in the comming years: recent

rapid development in high-resolution numerical modeling of the OML, including models of transport of suspended particles; and growing amounts of observational (*in situ* and remotely sensed) data from the polar regions of both hemispheres.

    Observations of nonuniform distribution of frazil on the sea surface have been available for several decades. Elongated zones of high ice concentration oriented roughly in the wind direction, so-called frazil streaks, have been described already in the 1980s. Eicken and Lange (1989) reported kilometer-long frazil streaks forming during katabatic wind events in a coastal

polynya in the Weddell Sea and, based on earlier works by Martin and Kauffman (1981) and Weeks and Ackley (1982), attributed their formation to Langmuir circulation. Today, thanks to satellite imagery (Fig. 1), it is known that frazil streaks are common in coastal polynyas of both hemispheres. They have been observed and analyzed, e.g., in Terra Nova Bay (Ciappa and

---

[c2] reference added
[c3] reference added
[c1] *Text added.*
[c2] ~~does have influence on~~

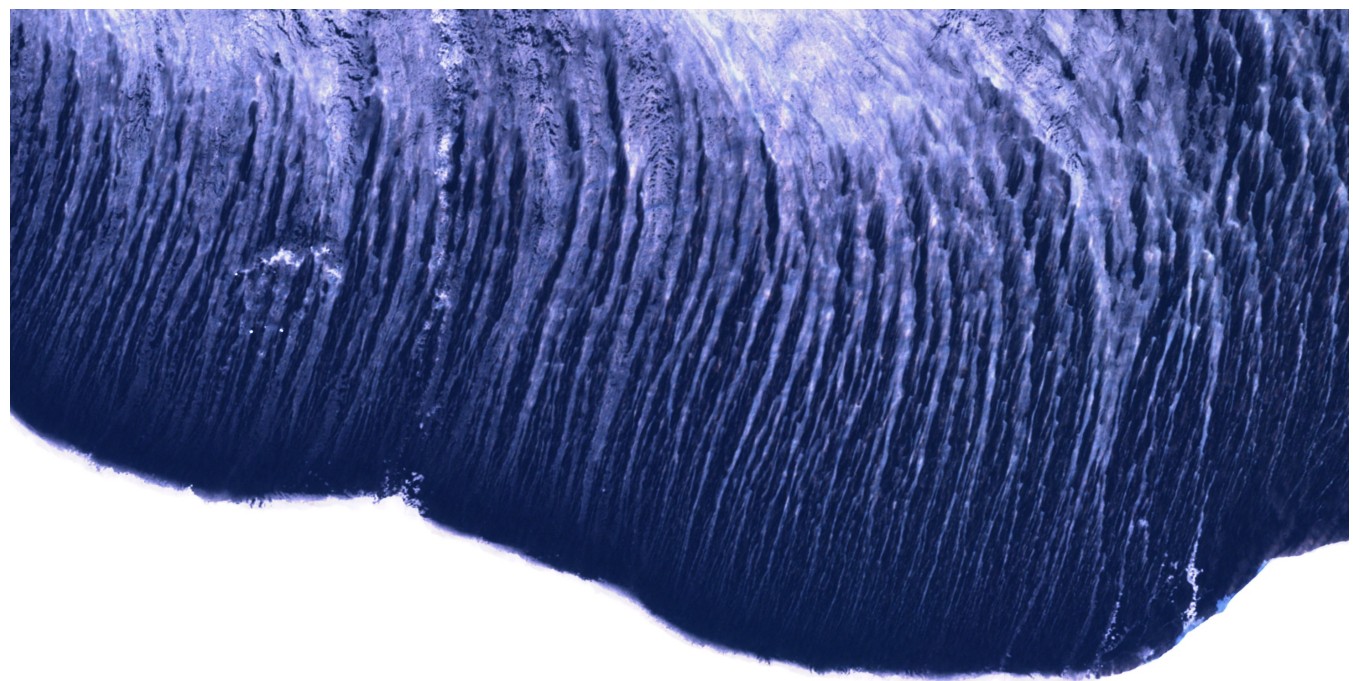

**Figure 1.** Sentinel-2 L1C image acquired on 10th March 2018, showing frazil streaks forming off the Ross Ice Shelf, Antractica (image size ca. 13×6 km, centered at 77.4°S, 173.3°E). Source: https://www.sentinel-hub.com/.

Pietranera, 2013; Hollands and Dierking, 2016; De Pace et al., 2019), Laptev Sea (Dmitrenko et al., 2010), and at the coasts of St. Lawrence Island (Drucker et al., 2003). Due to the three-dimensional character of the accompanying OML circulation, clouds of frazil crystals can be forced down the water column to depths of several tens of meters (Drucker et al., 2003; Ito et al., 2019), in shallower regions contributing to the entrainment of bottom sediments into the developing sea ice cover, and thus to sediment transport over large distances (Dethleff, 2005; Dethleff et al., 2009).

Since 1980s, frazil formation has been a subject of several numerical modelling studies. The recent, advanced models, in which a hydrodynamic model with evolution equations for temperature and salinity is coupled to transport equations describing the dynamics and thermodynamics of several crystal size classes (Holland and Feltham, 2005; Heorton et al., 2017; Rees Jones and Wells, 2018), are based on seminal works by Daly (1984)[c1], Omstedt and Svensson (1984) and Svensson and Omstedt (1994, 1998). Their models were one-dimensional, i.e., simulated the evolution of frazil volume fraction in a single water column, but included a whole range of mechanisms accompanying frazil formation, including the initial seeding, thermodynamic growth, flocculation, or secondary nucleation. Remarkably, although the new models enable fully three-dimensional (3D) simulations, they have been applied by their authors in very simplified settings – none of the studies cited above analyzed 3D aspects of frazil dynamics. Only recently, Matsumura and Ohshima (2015) used a Lagrangian frazil model coupled to a

---
[c1] reference added

hydrodynamic OML model to study the 3D structure of clouds of frazil crystals. They observed formation of frazil streaks in their model, although didn't analyze those features in any detail. Streak formation was analyzed earlier by Kämpf and Backhaus (1998, 1999), with a model in which frazil formed a layer on the sea surface, responding passively to convergence and divergence of surface currents, and with simple "toy models" by Thorpe (2009).

The goal of this paper is threefold. First, in view of the fact that the already mentioned rapid progress in large eddy simulation (LES) methods and research on particle-laden flows in the OML largely took place outside of the sea ice community, our aim is to provide the readers with a brief overview of the findings and concepts most important from the perspective of frazil modelling. We devote a separate section (2.1) to that review. The second goal is to analyze the 3D structure and variability of frazil clouds under different combinations of wind, wave and buoyancy forcing. And third, we address one of the largely

unanswered questions related to the frazil–OML interactions: whether frazil crystals can be regarded as positively buoyant, but otherwise passive tracers carried by the surrounding water, or whether they significantly modify the OML dynamics and properties, e.g., by weakening downwelling currents or reducing momentum and heat transfer from the atmosphere. In particular, we concentrate on the influence of frazil volume fraction on water density, heat and momentum transfer at the air–ocean interface, and the effective viscosity. Earlier studies indicate, e.g., that the net heat loss from the ocean to the atmosphere is

larger when ice is present within the water column as compared to analogous situations when it forms a layer at the sea surface (Matsumura and Ohshima, 2015). Laboratory observations and numerical simulations suggest that the turbulence level influences vertical distribution and size of frazil particles, and that in turn, layers of high ice volume fraction (grease ice) suppress turbulence (e.g., McFarlane et al., 2015; Clark and Doering, 2009; De Santi and Olla, 2017). [c1]Analogously, evolution of frazil and, especially, grease ice in the uppermost ocean layer is affected by waves and wave-generated turbulence, and wave energy

attenuation is modified by the volume fraction and thickness of the ice layer (e.g., Newyear and Martin, 1997; De la Rosa and Maus, 2012).[c2]

To reach the goals listed above, we implemented a frazil module in an open-source, non-hydrostatic hydrodynamic model dedicated to high-resolution simulations, we applied the coupled model to idealized simulations within a rectangular domain with periodic lateral boundaries, for varying atmospheric forcing, and, in a series of numerical experiments, we artificially

switched the selected frazil-related effects on or off in order to capture the role of those effects in shaping the OML circulation and bulk properties. In order to eliminate the complex feedbacks between frazil thermodynamics and dynamics and, especially, interactions between different size classes of frazil crystals, in the simulations analyzed in this paper all thermodynamic processes were artificially turned off. The behavior of the full model version will be studied in subsequent research.

The structure of the paper is as follows: in the next section, we introduce important concepts from modelling of OML

dynamics and material transport (section 2.1) and we discuss those concepts in the context of coastal polynyas and frazil formation in those water bodies (section 2.2). A summary of the hydrodynamic model and a detailed description of the frazil treatment are provided in section 3, followed by a description of the model configuration in section 4. The presentation of the

---

[c1] *Text added.*

[c2] references added

modelling results in section 5 progresses from the analysis of the general model behaviour under various forcing conditions to the model sensitivity to selected parameters and frazil–hydrodynamics coupling. We conclude with a discussion in section 6.

## 2 Mixing regimes in the OML in conditions favorable for frazil formation

### 2.1 Recent findings in research on OML dynamics and particle transport

As mentioned in the introduction, recent years have witnessed a rapid progress in two research areas important for the problems analyzed in this study: high-resolution, large-eddy-resolving numerical modeling of the ocean mixed layer; and modeling of transport of small particles within that layer. A full review of those developments is beyond the scope of this paper (see Chamecki et al., 2019, for an up-to-date review). Here, we only provide a brief summary of those aspects that are relevant for this study and for frazil modeling in general.

For freezing to occur within the ocean surface layer [c1](as opposed to freezing at the surface, leading to the formation of nilas) turbulent conditions are necessary. In general, turbulence generation might be associated with three forcing mechanisms: wind shear, waves, and [c2]buoyancy-driven convection. Based on [c3] Li et al. (2005), who proposed a two-dimensional (2D) phase diagram to classify different types of OML forcing, Belcher et al. (2012) defined three velocity scales, one for each turbulence generating mechanism, and introduced an improved phase diagram, based on those scales. They are: the surface friction velocity

$u_*$ for shear-driven turbulence, the vertical velocity $w_{*,L} = (u_S u_*^2)^{1/3}$ for wave-driven (i.e., Langmuir) turbulence (introduced by Grant and Belcher, 2009), and the convective velocity $w_* = (B_0 h)^{1/3}$ for buoyancy-driven turbulence (Deardorff, 1972). (In the definitions above, $u_S$ denotes the magnitude of the Stokes drift at the sea surface, $B_0$ the surface buoyancy flux, and $h$ the OML depth.) The diagram is constructed based on the ratios of those velocity scales (Fig. 2): the so-called turbulent Langmuir number $La_t = (u_*/u_S)^{1/2} = (u_*/w_{*,L})^{3/2}$, and $(w_*/w_{*,L})^3$, which corresponds also to the ratio of $h$ to the Langmuir stability

length $L_L$ [c4](an analogue of the Obukhov length, relevant for convective–Langmuir turbulence). Estimating the position of the analyzed situation in the $(La_t, w_*^3/w_{*,L}^3)$ plane enables to assign it to a certain mixing regime, i.e., to identify the dominating mechanisms of turbulent kinetic energy (TKE) dissipation. Notably, Belcher et al. (2012) showed that a mixed wave–buoyancy regime, with relatively small contribution from wind shear, dominates over most of the open oceans throughout the year. At the global scale, the probability density functions (pdfs) of $La_t$, estimated from reanalysis data, tend to be very narrow and peak

around $La_t = 0.3$ (thin vertical line in Fig. 2), which corresponds to fully developed seas. Higher values of $La_t$ are observed in regions/seasons with variable wind forcing and in semi-enclosed seas, i.e., in fetch-limited situations. This observation is relevant for frazil formation in coastal polynyas, in situations without swell, but with fetch-limited waves generated locally by strong katabatic winds. Conclusions from global studies, like that of Belcher et al. (2012), are hardly applicable to polynya and similar conditions. We will elaborate on those issues in section 2.2.

---

[c1] *Text added.*
[c2] ~~buoyancy (if it is negative, i.e., leads to convection)~~
[c3] ~~an earlier idea by~~
[c4] *Text added.*

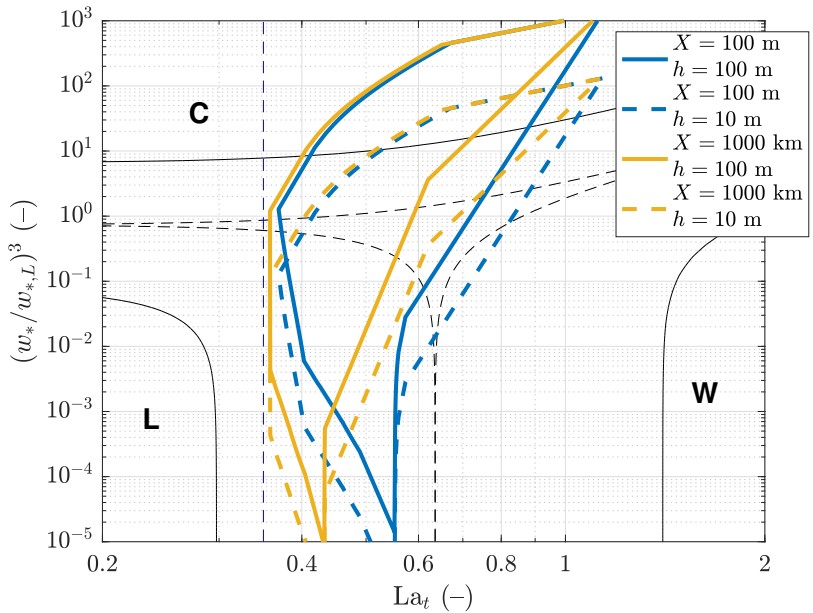

**Figure 2.** Regime diagram for OML mixing, following Belcher et al. (2012). Continuous black lines delineate regions where at least 90% of the total TKE is dissipated by only one of the three processes, Langmuir turbulence, wind shear, or convection (letters L, W and C, respectively). Similarly, dashed black lines mark regions where TKE dissipation due to one process exceeds 50% of the total dissipation. The thin dashed blue line is located at $\mathrm{La}_t = 0.35$, corresponding to fully developed waves. When the $(\mathrm{La}_t, w_*^3/w_{*,L}^3)$ pairs are computed for a range of wind, air temperature and wave conditions described in the text, the corresponding points fall into a region marked with a color contour, the exact position and shape of which depends on the assumed value of the mixed-layer depth $h$ (continuous lines: $h = 100$ m; dashed lines: $h = 10$ m) and fetch $X$ (blue: $X = 100$ m; yellow: $X = 1000$ km). See text for more details.

The OML dynamics under different mixing regimes has been analyzed numerically and theoretically in several studies (Chamecki et al., 2019). However, a great majority of those studies concentrates on non-convective situations, i.e., on combined shear- and wave-driven OMLs. Especially, large-eddy simulations with models including relevant wave-phase-averaged effects associated with the Stokes-drift velocity (the vortex force; the Coriolis–Stokes term; advection of water properties induced by wave-induced Lagrangian motion, etc.) have provided valuable insights into the nature of Langmuir circulation and transport.
5 Under highly idealized conditions, the average combined Ekman–Stokes flow can be computed analytically (McWilliams et al., 1997). In realistic, turbulent conditions, Langmuir "cells" rarely form regular, uniformly spaced and steady patterns, but rather have transient nature and tend to deform and merge with each other forming Y junctions pointing in downwind direction (Thorpe, 2004; Sullivan and McWilliams, 2010). Although several aspects of the unstable character of Langmuir circulation (or, rather, Langmuir turbulence) remain poorly understood, LES models successfully reproduce those features (McWilliams
10 et al., 1997; Chamecki et al., 2019, and references there). The low-order statistics of the OML flow and turbulence (mean profiles of velocity, velocity variance, TKE, TKE dissipation, etc.), as well as details of the 3D structure of the flow depend on several factors, including stratification, entrainment from the pycnocline (Grant and Belcher, 2009), misalignment between

the wind and wave direction (Van Roekel et al., 2012), the Ekman depth scale and thus the Coriolis parameter (Polton and Belcher, 2007), wave age and spectral characteristics (Harcourt and D'Asaro, 2008), or transient evolution of wind and wave forcing (Sullivan et al., 2012). As already mentioned, OML mixing under vertically unstable, i.e., convective conditions, has been less intensively studied (Deardorff, 1972; Moeng and Sullivan, 1994). Notably, whereas until very recently LES studies

were limited to idealized conditions and their results could rarely be validated with observations, LES methods are increasingly being applied to real ocean conditions with complex, spatially and temporally varying forcing (e.g., Fan et al., 2020).

Not surprisingly, the 3D structure of OML circulation and turbulence has essential influence on the transport and dispersion of particles in the upper ocean: oil droplets, gas bubbles, microplastics, sediment grains, phytoplankton, or ice crystals. Following Chamecki et al. (2019), those particles can be divided into passive tracers (following the motion of the surrounding water),

floaters (always remaining at the surface), sinking particles and buoyant particles (with density higher and lower, respectively, than the density of the surrounding water). In the great majority of studies, any inertial effects that might accompany the motion of the particles are ignored, so that the horizontal component of their velocity $\mathbf{u}_p = \mathbf{u}$ and the vertical component $w_p = w + w_t$, where $(\mathbf{u}, w)$ is the water velocity and $w_t$ is the terminal velocity of the particles. For the four particle types mentioned before, $w_t = 0$, $w_t \to \infty$, $w_t < 0$ and $w_t > 0$, respectively. Combined with the three velocity scales introduced earlier, $u_*$, $w_*$ and

$w_{*,L}$, the value of $w_t$ is crucial for the fate of the suspended material in the OML. Chor et al. (2018b), who analyzed OML transport of positively buoyant particles under pure convection, introduced a floatability parameter defined as a ratio $w_t/w_*$ and demonstrated that preferential concentration of particles in the surface convergence zones occurs for sufficiently high values of that parameter – particles with very low floatability are pulled down with the downwelling currents and resurface again in surface divergence regions, being effectively distributed over the entire OML, whereas particles with high floatability resurface

fast enough to stay trapped within the convergence zones. As $w_t$ depends on particles' density, size and shape, when multiple particle "classes" are present in the OML, smaller and/or denser ones tend to behave in the first, and larger/lighter ones in the second manner, producing a de-mixing effect. Apart from this basic mechanism generating preferential particle concentration in OML, another effect, relevant for objects with high $w_t/w_*$, leads to their accumulation in regions with high vorticity (Chor et al., 2018b).

Arguably, an analogous ratio of velocity scales can be defined for wave-forced OMLs, $w_t/w_{*,L}$, with similar influence on particle behaviour. This idea was expanded upon by Chor et al. (2018a), who proposed a combined velocity scale including all three TKE dissipation mechanisms. In turn, Yang et al. (2014) in their study on oil dispersion by Langmuir circulation use the ratio $u_S/w_t$, which they call drift-to-buoyancy parameter, i.e., they treat $u_S$ as a simple proxy of downwelling velocity in the OML circulation. Based on the results of LES simulations, Yang et al. (2014) show that the forms of the surface oil slicks

can be divided into three regimes, fingered, blurred and diluted, dependent on the value of $u_S/w_t$. In the fingered regime, for $u_S/w_t < 10$, oil droplets accumulate in windrows and are transported mostly in the wind/wave direction. In the diluted regime, for $u_S/w_t > 25$, the droplets are distributed over the entire OML, so that no clear structure of oil patches is visible and the net oil transport is directed at an angle to the wind direction (see also Yang et al., 2015). Thus, the floatability of suspended particles influences not only the vertical profiles of their mean concentration (as studied, e.g., by Kukulka et al., 2012; Yang

et al., 2014; Kukulka and Brunner, 2015; Chor et al., 2018a), but also their horizontal transport and diffusion (Yang et al., 2014, 2015).

Finally, it is worth stressing that in the majority of studies on material transport in the OML, including those cited above, an assumption is made that the volume concentration of suspended particles is sufficiently low so that the OML dynamics is not affected by their presence. The number of studies which explicitly concentrate on the influence of the transported material on OML behavior is rather limited. An example is the paper by Botte and Mansutti (2012), who used a LES model coupled to an ecosystem model to show that the influence of surfactants produced by phytoplankton on surface tension leads to formation of regions of high surface shear stress, to higher vertical velocities in the water column and deeper mixing.

## 2.2 Mixing regimes and frazil transport in [c1] polynyas

As mentioned in the introduction, most of the research on OML dynamics and material transport took place independently of research on frazil/grease ice formation – it is enough to say that, in the extensive review by Chamecki et al. (2019), cited earlier, the word "ice" is never used. Therefore, before proceeding to the details of the model and its setup, it is useful to take one more look at the issues referred to in the previous section, this time explicitly in the context of sea ice and conditions typical for its formation, with particular stress on polynyas, of interest in this work.

The three velocity scales characterizing OML TKE dissipation, $u_*$, $w_{*,L}$ and $w_*$, can be computed for a range of atmospheric and wave conditions that are likely to be found over coastal polynyas. Using the model of air–ocean heat and momentum exchange described in section 3.2 and in Supplementary Note S1, $u_*$ and $w_*$ can be computed for a given combination of wind speed $U_a$ and surface air temperature $T_a$, with just one additional, poorly constrained parameter: the mixed layer depth $h$. As the waves in polynyas are likely locally generated (i.e., no swell is present), the wave characteristics necessary to compute $w_{*,L}$ can be determined from wind speed and, again, one additional parameter: the fetch $X$. Details of the computation and definitions of the relevant variables can be found in Supplementary Note S2. The $(\mathrm{La}, w_*^3/w_{*,L}^3)$ pairs, necessary to determine the mixing regime in the diagram of Belcher et al. (2012), have been computed for different combinations of $U_a$ between 3 m/s and 30 m/s, $T_a$ between $-20°$C and $-1°$C, and for four possible combinations of two fetch values (100 m and 1000 km) and two mixed-layer depths (10 m and 100 m). The results are shown in Fig. 2. For the sake of legibility, no individual data points are plotted, just the boundary of regions surrounding them. As can be seen, in spite of large differences in $X$ and $h$, corresponding to very different conditions in terms of fetch (just offshore, at $X = 100$ m; and very far from shore, where waves become $X$-independent) and the vertical range of convection, the overall shape and position of the four regions remains relatively stable. All four regions are elongated in the $w_*^3/w_{*,L}^3$-direction, with their lower/upper parts corresponding to situations with small/large $|T_a - T_w|$ differences and thus weak/strong convective forcing ($T_w$ denotes surface water temperature, which is close to freezing point). A purely convection-dominated mixing regime (letter C in Fig. 2) requires not only $T_a \ll T_w$, but also very low wind speeds, i.e., conditions that are realistic only over short time periods, for example immediately after cessation of katabatic winds. Those conditions are conducive to ice formation in the form of nilas, with ice accumulation at the surface leading to a rapid decrease of ocean–atmosphere heat flux and eventually to the closing of the polynya.

---

[c1] latent-heat

At an another extreme, when $T_a \approx T_w$ (lower part of the diagram in Fig. 2), the dissipation results from combined effects of waves (Langmuir turbulence) and wind (vertical shear), which without the influence of swell are very closely related. Notably also, as the Stokes drift amplitude $u_S \sim a^2/T^3$, where $a$ is the wave amplitude and $T$ the wave period, the increase of $a$ with increasing fetch and wind speed is partially compensated by an accompanying increase of $T$ (see equations in Supplementary Note S2), which in effect reduces the range of values of $\mathrm{La}_t$ to between 0.35 and 0.55 (values $\mathrm{La} < 0.35$, to the left of the thin dashed blue line in Fig. 2, correspond to swell).

In general, the effectiveness of convective dissipation, and thus the contribution of convection to the total dissipation, increases with increasing $h$. Observations show that under typical conditions during strong katabatic wind events, the mixed layer thickness often reaches a few hundreds of meters. Shallow mixed layers are not realistic under those conditions unless, of course, the total water depth is small, as in some coastal polynyas in the Arctic (e.g., Dethleff, 2005; Ito et al., 2019).

Overall, a mixed TKE dissipation regime should be typical for coastal polynyas, with non-negligible contributions from all three mixing mechanisms, but with the domination of convective mixing or Langmuir turbulence under very strong or relatively weak atmospheric cooling, respectively. Hence, all three mechanisms should be captured by any model intended to reproduce the OML behaviour in those water bodies. Importantly also, when sea ice formation takes place, it introduces an additional source of buoyancy, related to salt rejection to the water column – an effect that is not included in the hitherto discussion and that may additionally increase the relative role of convective processes.

As elaborated upon in the previous section, the fate of the forming frazil ice crystals in the turbulent OML depends largely on their rising velocity $w_t$ in relation to the three OML velocity scales. For the conditions analyzed here, $u_*$, $w_{*,L}$ and $w_*$ reach 0.05, 0.1 and 0.02 m/s, respectively (Supplementary Figs. 3 and 5). Anticipating the presentation of the results in section 5, it is worth adding that instantaneous vertical velocity in downwelling plumes can be much higher and exceed 0.3–0.4 m/s. The terminal velocity of frazil crystals, computed from their radius and density (e.g., Holland and Feltham, 2005) is lower than 0.001 m/s for crystals smaller than 0.1 mm and increases to ~0.01 m/s for crystals with radius >1 mm. Thus, the floatability of frazil in the conditions of interest is low and <1 for most crystal size classes. Based on this elementary analysis alone, preferential concentration of frazil within the turbulent OML [c1]can be expected to occur only for the largest crystals.

## 3  Model description

The model code developed for this study is based on the Coastal and Regional Ocean COmmunity Model (CROCO), which is built upon the Regional Ocean Modeling System (ROMS; Shchepetkin and McWilliams, 2005). CROCO is an open-source hydrodynamic model, developed for high-resolution, non-hydrostatic and non-Boussinesq simulations of various flow types, especially at fine spatial and temporal scales. Examples of applications include coastal upwelling systems, frontal dynamics, internal waves, and three-dimensional simulations of wave–current interactions in the coastal zone, as well as selected biogeochemical processes and sediment transport. In section 3.1, we provide a short summary of the model features relevant for the setup used in this study; a detailed description of CROCO can be found in the documentation available through the model web

---

[c1] should be limited to the largest crystals

page (http://www.croco-ocean.org). In the subsequent sections 3.2 and 3.3, an in-depth description of new model features can be found, including details of the frazil module.

## 3.1 The CROCO model

The equations of CROCO are formulated in terms of the following wave-averaged variables: dynamic pressure $p_c = p + p_w$, sea level $\xi$, density $\rho$, and velocity $(\mathbf{u}_c, w_c) = (\mathbf{u}, w) + (\mathbf{u}_S, w_S)$, where $p_w$ denotes the so-called Bernoulli head, related to the wave-averaged kinetic energy, and $(\mathbf{u}_S, w_S)$ denotes the Stokes velocity, related to the wave motion. Thus, the total velocity $(\mathbf{u}_c, w_c)$ is a Lagrangian, wave-averaged velocity, given as a sum of the Eulerian and Stokes component. In the notation used above, $\mathbf{u}_c$, $\mathbf{u}$, $\mathbf{u}_S$ denote the horizontal components, and $w_c$, $w$, $w_S$ the vertical components of the respective vectors. The governing equations are [c1]the momentum equations in the horizontal and vertical direction, the continuity equation and the surface kinematic condition:

$$\frac{\partial(\rho\mathbf{u})}{\partial t} + \nabla \cdot (\rho\mathbf{u}_c\mathbf{u}) + \frac{\partial}{\partial z}(\rho w_c\mathbf{u}) + f\rho\mathbf{n}_z \times \mathbf{u}_c = -\nabla p_c + \rho\mathbf{u}_S\nabla \cdot \mathbf{u} + \mathbf{F}_h + \mathbf{F}_w, \tag{1}$$

$$\frac{\partial(\rho w)}{\partial t} + \nabla \cdot (\rho\mathbf{u_c}w) + \frac{\partial}{\partial z}(\rho w_c w) = -\frac{\partial p_c}{\partial z} - \rho g + \rho\mathbf{u}_S \cdot \frac{\partial \mathbf{u}}{\partial z} + F_v, \tag{2}$$

$$\frac{\partial \rho}{\partial t} + \nabla \cdot (\rho\mathbf{u}) + \frac{\partial}{\partial z}(\rho w_c) = 0, \tag{3}$$

$$\frac{\partial \xi}{\partial t} + \mathbf{u} \cdot \nabla\xi = w_\xi, \tag{4}$$

where $\nabla$ is the horizontal gradient operator, $f = 2\Omega_Z \sin\phi$ denotes the Coriolis parameter (with $\Omega_Z$ the angular velocity of the Earth and $\phi$ the latitude), $g$ denotes acceleration due to gravity, $\mathbf{n}_z$ is a unit vector directed vertically upward, $w_\xi$ is the vertical velocity at the surface, $\mathbf{F}_w$ denotes wave-related non-conservative forces (e.g., due to wave breaking) and $(\mathbf{F}_h, F_v)$ denotes non-conservative forces unrelated to waves (e.g., subgrid-scale turbulence). The treatment of the non-hydrostatic terms, including the relationship between pressure and density, needed to close the system of equations (1)–(4), are provided in the documentation of CROCO, cited above. Theoretical background underlying wave-related terms in (1)–(4), together with derivation, boundary conditions and discussion of the influence of individual terms on the mixed layer dynamics, can be found in McWilliams et al. (2004) and Uchiyama et al. (2010).

Equations (1)–(4) are supplemented by a transport equation for any scalar quantity $\alpha$:

$$\frac{\partial(\rho\alpha)}{\partial t} + \nabla \cdot (\rho\mathbf{u}_c\alpha) + \frac{\partial}{\partial z}(\rho w_c\alpha) = S_\alpha + S_{\alpha,w}, \tag{5}$$

where $S_\alpha$ and $S_{\alpha,w}$ denote the wave-unrelated and wave-related, respectively, non-conservative (forcing and diffusive) terms. The formulation of those terms depends on the type of "tracer" $\alpha$. In particular, in the case of both temperature $T$ and salinity $S$, the source terms include turbulent diffusion, as well as forcing from the atmosphere at the ocean surface (see further). In the case of frazil crystals, equation (5) is solved for the volume fraction of each frazil size class, with source terms representing thermodynamic processes (not used in this study) and buoyancy, as described in detail in Section 3.3.

---

[c1] *Text added.*

To solve the system of equations (1)–(5), the equation of state is necessary, relating the density of water $\rho_w$ to temperature and salinity, $\rho_w = \rho_w(T, S)$. As in previous models of frazil processes (e.g., Holland and Feltham, 2005), a linearized equation of state is used:

$$\rho_w = \rho_{w,0}[1 + \beta_T(T - T_0) + \beta_S(S - S_0)], \tag{6}$$

with $\rho_{w,0} = \rho_{w,0}(T_0, S_0) = 1028.15\,\text{kg}\cdot\text{m}^{-3}$, $T_0 = -1°\text{C}$, $S_0 = 35\,\text{PSU}$, $\beta_T = 4\cdot10^{-5}\,\text{K}^{-1}$, $\beta_S = 7.8\cdot10^{-4}\,\text{PSU}^{-1}$. In equations (1)–(5), $\rho = \rho_w$ if no ice is present or its influence on density is switched off; otherwise $\rho$ represents the density of the ice–water mixture, as described in section 3.3.3.

In CROCO, the wave-related variables necessary to compute the wave-related terms in the governing equations can be obtained from a spectral wave model, coupled to the hydrodynamic model. In this study, the model is run on a relatively small, rectangular domain with periodic lateral boundaries, which implies conditions that are spatially uniform at the domain scale. Accordingly, it is assumed that the phase-averaged wave characteristics – significant wave height $H_s$, peak wave period $T_p$, etc. – are constant and depend exclusively on the assumed wind speed $U_a$ and fetch $X$, i.e., that the wave field is stationary and fetch-limited. As described in Supplementary Note S2, a simple empirical model is used to compute $H_s(U_a, X)$ and $T_p(U_a, X)$. From those wave characteristics, the Stokes drift and other wave-related variables are obtained, assuming monochromatic, non-breaking, deep-water waves. In particular, the amplitude of the Stokes drift at the sea surface, $z = 0$, is given as:

$$u_S = \omega_p k_p a^2, \tag{7}$$

where $\omega_p$ is the peak wave frequency, $k_p$ is the corresponding wavenumber (in deep water, $k_p = \omega_p^2/g = 4\pi^2 g/T_p^2$) and $a = H_s/2$ the wave amplitude.

The non-conservative forcing terms in equations (1), (2) and (5) include turbulent viscosity/diffusion terms, $\mathbf{F}_{t,h}$, $F_{t,v}$ and $S_{\alpha,t}$, respectively. The Reynolds stresses and turbulent tracer fluxes required to compute those terms are:

$$\overline{\mathbf{u}'w'} = -K_M \frac{\partial \overline{\mathbf{u}}}{\partial z} \quad \text{and} \quad \overline{\alpha'w'} = -K_\alpha \frac{\partial \overline{\alpha}}{\partial z}, \tag{8}$$

where $\mathbf{u}'$, $w'$, $\alpha'$ are turbulent fluctuations about the slowly-varying values, denoted with overbars, and the eddy viscosity $K_M$ and eddy diffusivity $K_\alpha$ have the form:

$$K_M = c\sqrt{2k}lS_M + \nu \quad \text{and} \quad K_\alpha = c\sqrt{2k}lS_\alpha + \nu_\alpha, \tag{9}$$

where $\nu$ and $\nu_\alpha$ denote the molecular viscosity and diffusivity, respectively, $S_M$ and $S_\alpha$ are stability functions, describing the effects of stratification and shear, and the coefficient $c$ depends on the form of $S_M$ and $S_\alpha$. The turbulent kinetic energy (TKE) per unit mass $k$, and the length scale $l$ are determined with the so-called generic length scale (GLS) method developed by Umlauf and Burchard (2003) and implemented in ROMS, and later in CROCO, by Warner et al. (2005). The advantage of GLS is that it provides a unified framework for a number of popular two-equation turbulence models, taking advantage of the similarities in their mathematical formulations. The GLS scheme solves the transport equation for $k$ (common to all

models) and the equation for a generic variable $\psi$, used to determine the length scale $l$ from a relationship $\psi = (c_\mu^0)^p k^m l^n$. The equations have the form:

$$\frac{\partial k}{\partial t} + \overline{\mathbf{u}} \cdot \nabla k = \frac{\partial}{\partial z}\left(\frac{K_M}{\sigma_k}\frac{\partial k}{\partial z}\right) + P + B - \varepsilon, \tag{10}$$

$$\frac{\partial \psi}{\partial t} + \overline{\mathbf{u}} \cdot \nabla \psi = \frac{\partial}{\partial z}\left(\frac{K_M}{\sigma_\psi}\frac{\partial \psi}{\partial z}\right) + \frac{\psi}{k}(c_1 P + c_3 B - c_2 \varepsilon F_{\text{wall}}), \tag{11}$$

where the coefficients $c_\mu^0$, $c_1$, $c_2$, $c_3$, $\sigma_k$, $\sigma_\psi$, the wall proximity function $F_{\text{wall}}$, as well as the exponents $p$, $m$, $n$ are model-dependent. The terms $P$ and $B$ denote production of $k$ due to shear and buoyancy, respectively, and $\varepsilon$ represents TKE dissipation (see Warner et al., 2005, for details and formulation of those terms). Importantly, all those parameters can be adjusted to obtain the standard $k$–$\varepsilon$, $k$–$\omega$ and $k$–$kl$ models. In the simulations described in this paper, the $k$–$\varepsilon$ GLS version was used (for which $p = 3.0$, $m = 1.5$, $n = -1.0$), with stability functions according to Canuto et al. (2001).

As the molecular diffusivity is much lower than the turbulent one in all situations considered here ($O(10^{-7})$ m$^2 \cdot$s$^{-1}$ for heat, $O(10^{-9})$ m$^2 \cdot$s$^{-1}$ for salt), all $\nu_\alpha$ are assumed equal to zero. In the case of the viscosity $\nu$, its value corresponds to the viscosity of water $\nu_w$ is no frazil is present or its influence is switched off; otherwise it is computed as described in section 3.3.3.

The horizontal mixing is computed with Smagorinsky parametrization of lateral turbulent viscosity, with a constant Prandtl number of 0.3.

## 3.2  Forcing from the atmosphere

In the simplified model setting considered here, the state of the lower atmosphere is described by the following set of variables: wind speed vector at 10-m height $\mathbf{U}_a$, air temperature and relative humidity, $T_a$ and $q_r$, respectively, and surface atmospheric pressure $p_a$. The shortwave radiation flux and the precipitation rate are assumed equal to zero. The net heat flux at the ocean–atmosphere interface is computed as the sum of the turbulent latent and sensible heat flux, $F_h$ and $F_e$, and the sum of the upward and downward long-wave radiation, $F_{\text{rad}}$. All flux components, as well as the wind stress $\tau_w$ and the surface salt flux due to evaporation $H_e$ are computed using modified bulk formulae from COAMPS (Coupled Ocean/Atmosphere Mesoscale Prediction System). They are described in the Supplementary Note S2. The major difference with respect to the analogous formulae available in the public version of CROCO is related to the computation of the transfer coefficients for heat, moisture and momentum, which in CROCO are assumed constant. Here, in order to make them suitable for a wide range of conditions, including very high wind speeds, and, even more importantly, to allow for transfer coefficients varying with surface frazil volume fraction, they are variable. The general formulae are:

$$C_d = C_{d,n}\alpha_s, \quad C_h = C_{h,n}\alpha_s, \quad C_e = C_{e,n}\alpha_s, \tag{12}$$

where $C_{x,n} = C_{x,n}(U_a)$ are neutral transfer coefficients (with $x$ denoting [c1] '$d$', '$h$' or '$e$' for transfer of momentum, heat and water, respectively), $\alpha_s = \alpha_s(U_a, T_w - T_a)$ is a stability parameter, and $U_a = |\mathbf{U}_a|$.

---

[c1] '$d$', '$h$' or '$e$'

### 3.3 Frazil processes

#### 3.3.1 Definitions and assumptions

As in most similar models (e.g., Svensson and Omstedt, 1998; Holland and Feltham, 2005; Heorton et al., 2017), frazil is represented by $i = 1, \ldots, N_f$ prescribed crystal size classes. It is assumed that the crystals are disk shaped, so that their size in the $i$th class can be described by the radius $r_i$ and thickness $h_i = a_{r,i} r_i$. In this study, [c1]following Holland and Feltham (2005)[c2], the aspect ratio $a_{r,i}$ is assumed the same for each class, $a_r = 1/50$ (see Table 1). Similarly, a constant ice density is assumed, $\rho_i = 920$ kg·m$^{-3}$. The amount of frazil from each size class in a given volume of sea water is described by ice volume fraction $c_i$ ($0 \leq c_i \leq 1$). The total frazil volume fraction $c = \sum_{i=1}^{N_f} c_i$.

As mentioned in the introduction, the thermodynamic model of frazil has been switched off in all simulations described in this paper. Consequently, the total, domain-integrated mass of ice within each size class remains constant, equal to its initial value (see further section 4). The OML-averaged volume fraction of the $i$th class, $\tilde{c}_i$, is defined as $\tilde{c}_i = h^{-1} \int_{z=-h}^{0} c_i dz$.

#### 3.3.2 Frazil transport

The transport of frazil crystals within the OML, as for every scalar quantity, is described by equation (5). [c3]Zero-flux boundary conditions are used at the surface and at the bottom of the model domain. In the full model version, including ice thermodynamics, the source terms in [c4] equation (5) include those related to freezing/melting, secondary nucleation, etc. With those processes switched off, the only source term is related to the positive buoyancy of crystals and their precipitation towards the water surface. For each frazil size class $i$:

$$S_i = -\frac{\partial}{\partial z}(w_{t,i} c_i), \tag{13}$$

where $w_{t,i}$ denotes the vertical velocity of ice relative to the surrounding water. In the uppermost layer of the model $w_{t,i}(z = 0) = 0$. Elsewhere in the water column, disregarding inertial effects, $w_{t,i}$ can be approximated with the terminal velocity of crystals, computed from the balance between the gravity and buoyancy force:

$$\pi a_{r,i} r_i^3 (\rho_w - \rho_i) g = \frac{1}{2} C_D \rho_w \pi r_i^2 w_{t,i}^2, \tag{14}$$

so that:

$$2 a_{r,i} \frac{\rho_w - \rho_i}{\rho_w} g = \frac{C_{D,i} w_{t,i}^2}{r_i}. \tag{15}$$

---

[c1] *Text added.*
[c2] reference added
[c3] *Text added.*
[c4] ~~that~~

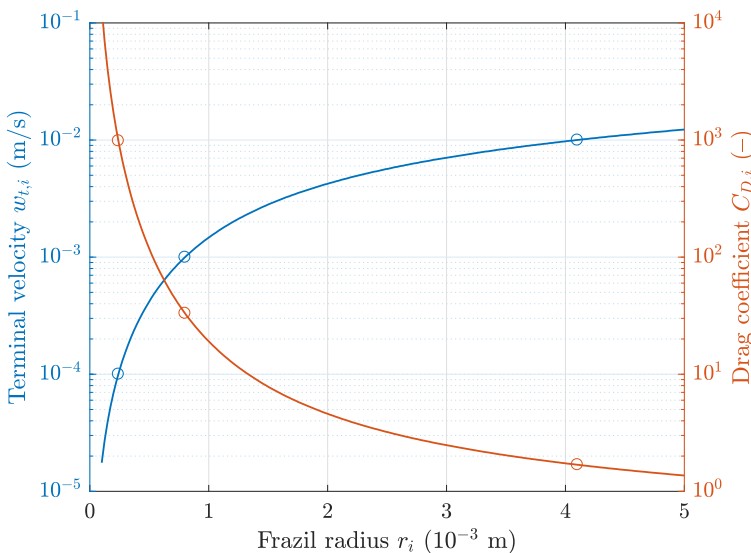

**Figure 3.** Terminal velocity $w_{t,i}$ (left axis) and drag coefficient $C_{D,i}$ (right axis) computed from formulae (15)–(17) for frazil radii $r_i$ between 0.1 and 5.0 mm. Circles mark the values used in the simulations analyzed in this study (see Table 1).

The widely used Schiller–Naumann model (Kolev, 2011)[c5] is used to calculate the drag coefficient $C_{D,i}$:

$$C_{D,i} = \begin{cases} \dfrac{24}{\mathrm{Re}_i}\left(1+0.15\mathrm{Re}_i^{0.687}\right) & \mathrm{Re}_i < 1000, \\ 0.44 & \mathrm{Re}_i \geq 1000, \end{cases} \tag{16}$$

where the relative (or particle) Reynolds number $\mathrm{Re}_i$:

$$\mathrm{Re}_i = \frac{2r_i w_{t,i}}{\nu_w}. \tag{17}$$

5  For realistic values of $r_i$, $\mathrm{Re}_i \ll 1000$, so that the first expression in (16) is relevant. The values of $w_{t,i}$ and $C_{D,i}$ obtained from (15)–(17) for a range of frazil radii $r_i$ is shown in Fig. 3. [c1]Notably, although simple, this model is in broad agreement with observed relationships between frazil crystal size and rise velocity, as well as with the existing theoretical models (see McFarlane et al., 2014, and references there)[c2]. [c3]When necessary, equations (15)–(17) [c4]can be easily replaced with more complex ones, e.g., with models taking into account crystal orientation (in the present study, concentrating on turbulent conditions,

10  the assumption of random orientation seems justified).

---

[c5] reference added
[c1] *Text added.*
[c2] reference added
[c3] *Text added.*
[c4] *Text added.*

### 3.3.3 Frazil influence on hydrodynamics

In the simulations in section 5, the presence of frazil is allowed to modify the OML dynamics by influencing four variables: the net density $\rho$, the net neutral transfer coefficients at the ocean–atmosphere interface $\hat{C}_{x,n}$, the effective viscosity $\nu$, and the drag force $F_D$ exerted by the rising frazil crystals on the surrounding water. The density of the water–ice mixture is:

$$\rho = (1-c)\rho_w + c\rho_i. \tag{18}$$

For the transfer coefficients, a linear relationship is assumed for frazil volume fractions between $c = 0$ and $c = c_{\mathrm{lim}}$, and a constant value $C_{x,n,\mathrm{lim}}$ for $c > c_{\mathrm{lim}}$:

$$\hat{C}_{x,n} = \max\{C_{x,n} + (C_{x,n,\mathrm{lim}} - C_{x,n})c/c_{\mathrm{lim}}, C_{x,n,\mathrm{lim}}\}, \tag{19}$$

where $C_{x,n}$ is computed from formulae (12). We assume $C_{x,\mathrm{lim}} < C_x$, i.e., in agreement with observations, the presence of frazil/grease ice leads to a decrease of the ocean–atmosphere heat and momentum exchange (contrary to other sea ice types, for which, in the case of momentum transfer, the effect is opposite). The roughness length of grease ice equals $2.7 \cdot 10^{-6}$ m (as compared to, e.g., $1.0 \cdot 10^{-3}$ m for pancake ice and $1.3$–$2.4 \cdot 10^{-3}$ m for first-year ice) and its neutral drag coefficient is as low as $0.7 \cdot 10^{-3}$ (Guest and Davidson, 1991).

Several models have been used in the literature to compute the effective viscosity $\nu$ of the water–ice mixture. Here, following Matsumura and Ohshima (2015), a simple expression is used:

$$\nu = \nu_w \left[1 + 2.5c_{\mathrm{rel}} + 10.05c_{\mathrm{rel}}^2 + 2.73 \cdot 10^{-3}\exp(16.6c_{\mathrm{rel}})\right], \tag{20}$$

in which $c_{\mathrm{rel}} = \min\{c/c_{\max}, 1.0\}$ and $c_{\max}$ is the maximum ice volume fraction, corresponding to typical values in grease ice. For $c_{\mathrm{rel}} = 0$, $\nu = \nu_w$; for $c_{\mathrm{rel}} = 1$, $\nu = 4 \cdot 10^4 \nu_w$, which is in agreement with observations (e.g., Martin and Kauffman, 1981; Newyear and Martin, 1997). More advanced models for $\nu$ have been proposed (e.g., De Carolis et al., 2005), but their range of applicability is often limited to a narrow range of conditions (e.g., very low ice concentration). For the present, idealized modeling study, formula (20) is sufficient to capture the leading-order effects related to the $\nu(c)$ variability.

Following Covello et al. (2016), the net liquid–solid drag force for a single dispersed phase can be computed from:

$$F_{D,i} = \begin{cases} -\dfrac{3\rho_w C_{D,i} w_{t,i}^2}{8r_i}\dfrac{c_i}{(1-c_i)^{1.65}} & c_i \leq 0.2, \\[2ex] -\dfrac{75\rho_w \nu_w w_{t,i}}{2r_i^2}\dfrac{c_i}{1-c_i} - \dfrac{7\rho_w w_{t,i}^2}{16r_i}c_i & c_i > 0.2. \end{cases} \tag{21}$$

Importantly, this formulation is consistent with formulae (15)–(17) used to compute the terminal velocity $w_{t,i}$. Moreover, for $c_i \leq 0.2$ – which is true for the great majority of cases analyzed in this study – it follows from (15) that the term $\rho_w C_{D,i} w_{t,i}^2/r_i$ is constant, equal to $2a_r(\rho_w - \rho_i)g$ (black curve in Fig. 4). Therefore, the net $F_D$ is computed from the total ice volume fraction:

$$F_D = \begin{cases} -\dfrac{3a_r(\rho_w - \rho_i)g}{4}\dfrac{c}{(1-c)^{1.65}} & c \leq 0.2, \\[2ex] -\dfrac{75\rho_w \nu_w \tilde{w}_t}{2\tilde{r}^2}\dfrac{c}{1-c} - \dfrac{7\rho_w \tilde{w}_t^2}{16\tilde{r}}c & c > 0.2, \end{cases} \tag{22}$$

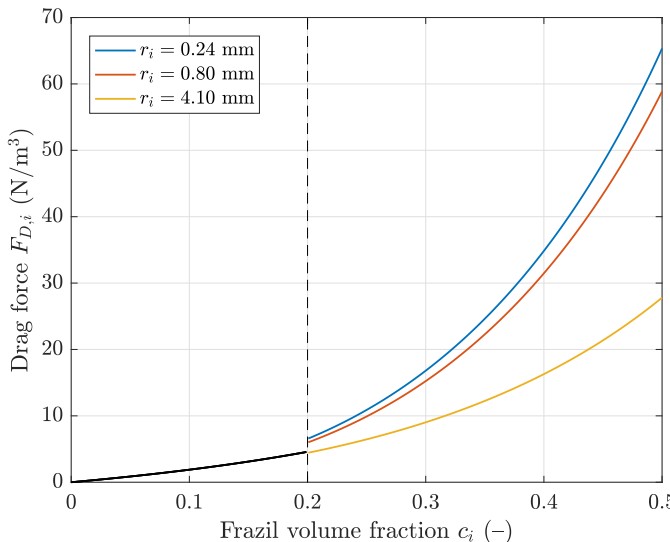

**Figure 4.** Drag force $F_{D,i}$ induced by the vertical motion of frazil crystals in function of frazil volume fraction $c_i$. For $c_i < 0.2$, $F_{D,i}$ is frazil-size-independent (black curve). For $c_i > 0.2$, three curves are shown, corresponding to the three values of frazil radius $r_i$ used in the simulations discussed in this paper (see also circle symbols in Fig. 3).

where $\tilde{w}_t$ and $\tilde{r}$ denote volume-fraction-weighted average rising speed and radius, respectively.

[c1]Crucially, although the modified version of CROCO includes the frazil-induced effects described by equations (18)–(22), [c2]the general form of the governing equations, described in section 3.1, remained unaffected. Thus, formally the model is only valid for total ice volume fraction $c \ll 1$, not for a mixture of water and ice with any $c \leq 1$, and care should be taken that the

5   condition of small $c$ is not violated during model applications.

### 3.4   Model testing

No validation against observational data has been performed in this study. In order to test the suitability of the configuration of CROCO described above to the conditions of interest, the model has been set up to reproduce the test case analyzed by McWilliams et al. (1997) and later by Yang et al. (2015), with reference to observational data by D'Asaro (2001). The test

10  case is in several aspects very similar to the setup considered here. The results are presented and discussed in Supplementary Note S3. Additionally, qualitative comparison of our results has been made with the results of Chen et al. (2016) and Chor et al. (2018b). We return to the issue of validation of the modeling results in the discussion.

---

[c1] *Text added.*

[c2] *Text added.*

## 4 Model configuration

In this work, an idealized model setup is used. The model domain is cuboid, placed at the latitude $\phi = 75°$N, with periodic lateral boundaries in both horizontal directions, covering the upper 150 m of the ocean within an area of $1200 \times 1200$ m$^2$. The horizontal resolution $\Delta x = 3$ m, and there are 60 nonuniformly distributed layers, so that the vertical resolution $\Delta z$ varies from 0.5 m at the surface to 9.2 m at the lower boundary. A summary of all model parameters is given in Table 1.

The model is initialized with spatially uniform salinity and temperature profiles, with the initial mixed layer depth $h = 100$ m and:

$$T(z) = \delta T + \begin{cases} -1.5 & z \geq -h, \\ -0.5 & z < -h, \end{cases} \tag{23}$$

$$S(z) = \delta S + \begin{cases} 34.0 & z \geq -h, \\ 35.0 & z < -h, \end{cases} \tag{24}$$

where $\delta S(x,y,z)$ and $\delta T(x,y,z)$ are random numbers drawn from a uniform distribution over an interval $(-0.005, 0.005)$ PSU and $(-0.025, 0.025)°$C, respectively, added to the profiles in order to introduce an initial disturbance to the otherwise perfectly uniform conditions. The water velocity is initialized with a horizontally uniform, steady-state Stokes–Ekman solution (McWilliams et al., 1997) in order to reduce the model spin up time. A free-slip boundary condition is applied at the lower boundary.

In simulations with frazil thermodynamics, it is important that the number of frazil size classes $N_f$ is sufficiently large, so that the crystal size distribution and interactions between different classes can be correctly captured. In this study, with frazil thermodynamics switched off, $N_f$ is set to three, and the radii of crystals are selected so that their terminal velocity equals $10^{-4}$ m·s$^{-1}$ (for $r_1 = 0.24$ mm), $10^{-3}$ m·s$^{-1}$ (for $r_2 = 0.80$ mm) and $10^{-2}$ m·s$^{-1}$ (for $r_3 = 4.10$ mm), see circle symbols in Fig. 3. For all $i$, the same OML-average volume fraction is set, $\tilde{c}_i = 0.00168$, which corresponds to a layer of ice of thickness 0.168 m for each class and the total ice thickness of 0.5 m. At time $t = 0$, frazil is uniformly distributed over the upper $0.8h$ meters of the model domain (tests have shown that [c1]the initial spatial distribution of frazil does not influence the final results).

The simulations were performed for different combinations of the following variable parameters: the air temperature $T_a$ and wind speed $U_a$. [c2]The wind and wave direction is along the $x$-axis. The corresponding wave forcing, net heat flux from the atmosphere, wind stress, and the velocity scales $u_*$, $w_*$ and $w_{*,L}$ are listed in Table 2. The wind fetch $X$ was set to 100 km. Some tests with very small $X$, equal to 1000 m, have been performed, but are not discussed here, as their results do not alter the conclusions from model runs with large $X$ (although, obviously, there are quantitative differences in the modelled velocity profiles and other characteristics). Test model runs with $h < 100$ m have been performed as well, but very fast deepening of the OML was observed due to strong mixing, so that the final results were close to those obtained with $h = 100$ m.

For each combination of $T_a$ and $U_a$, six series of simulations were run (Table 3), differing in terms of coupling between the frazil and hydrodynamic parts of the model: from $\mathcal{F}_0$ in which the hydrodynamics is unaffected by the presence of frazil (and

---

[c1] ~~this choice~~

[c2] *Text added.*

**Table 1.** Setup of the CROCO model used in this study

| Description | Value |
|---|---|
| ***Physical constants/prescribed model parameters*** | |
| Acceleration due to gravity $g$ | 9.81 m·s$^{-1}$ |
| Specific heat of water $c_{p,w}$ | 3985 J·kg$^{-1}$·K$^{-1}$ |
| Specific heat of air $c_{p,a}$ | 1004.8 J·kg$^{-1}$·K$^{-1}$ |
| Stefan–Boltzmann constant $\sigma_{\mathrm{SB}}$ | 5.6697·10$^{-8}$ W·m$^{-2}$·K$^{-4}$ |
| Reference density of sea water $\rho_{w,0}$ | 1028.15 kg·m$^{-3}$ |
| Reference water temperature $T_0$ | $-1°$C |
| Reference salinity $S_0$ | 35 PSU |
| Thermal expansion coefficient of sea water $\beta_T$ | 4·10$^{-5}$ K$^{-1}$ |
| Saline contraction coefficient $\beta_S$ | 7.8·10$^{-4}$ PSU$^{-1}$ |
| Molecular viscosity of water $\nu_w$ | 1.95·10$^{-6}$ m$^2$·s$-1$ |
| Angular velocity of the Earth $\Omega_Z$ | 7.29·10$^{-5}$ s$^{-1}$ |
| Ice density $\rho_i$ | 920 kg·m$^{-3}$ |
| Aspect ratio of frazil crystals $a_r$ | 0.02 |
| Number of frazil size classes $N_f$ | 3 |
| Radius of frazil crystals $r_i$ | 0.24, 0.80, 4.10 mm |
| Relative humidity of the air $r_{rel}$ | 1.0 |
| Sea level pressure $p_a$ | 1013 hPa |
| Wind fetch $X$ | 100 km |
| ***Model domain and numerical settings*** | |
| Domain size $L_x \times L_y \times L_z$ | 1200×1200×150 m |
| Latitude $\phi$ | 75°N |
| Initial mixed layer depth $h$ | 100 m |
| Horizontal resolution $\Delta x$ | 3 m |
| Vertical resolution $\Delta z$ | 0.5–9.2 m |
| Number of layers $n_z$ | 60 |
| Time step $\Delta t$ | 0.25 s |
| ***Variable parameters*** | |
| Air temperature $T_a$ | $-20°$C, $-1.5°$C |
| Wind speed $U_a$ | 5, 15, 30 m·s$^{-1}$ |

**Table 2.** Model forcing [c3]computed for different combinations of $U_a$ and $T_a$ [c4]with formulae described in Supplementary Note S1

| $U_a$ | $H_s$ | $T_p$ | $u_S$ | $\tau_w$ | $F_{net}$ | $w_*$ | $u_*$ | $w_{*,L}$ | $La_t$ | $w_*^3/w_{*,L}^3$ |
|---|---|---|---|---|---|---|---|---|---|---|
| m/s | m | s | m/s | N·m$^{-2}$ | W·m$^{-2}$ | m/s | m/s | m/s | – | – |
| | | | | $T_a = -20°C$ | | | | | | |
| 5.0 | 0.52 | 3.41 | 0.04 | 0.06 | $-393$ | 0.014 | 0.008 | 0.014 | 0.43 | 1.07 |
| 15.0 | 2.86 | 7.55 | 0.12 | 0.58 | $-850$ | 0.018 | 0.024 | 0.041 | 0.45 | 0.09 |
| 30.0 | 7.12 | 11.24 | 0.23 | 2.95 | $-1459$ | 0.022 | 0.050 | 0.082 | 0.47 | 0.02 |
| | | | | $T_a = -1.5°C$ | | | | | | |
| 5.0 | 0.52 | 3.41 | 0.04 | 0.04 | 1 | 0.001 | 0.006 | 0.012 | 0.38 | $10^{-3}$ |
| 15.0 | 2.86 | 7.55 | 0.12 | 0.46 | 4 | 0.002 | 0.021 | 0.038 | 0.42 | $10^{-4}$ |
| 30.0 | 7.12 | 11.24 | 0.23 | 2.55 | 8 | 0.002 | 0.046 | 0.079 | 0.45 | $10^{-5}$ |

**Table 3.** Main series of simulations discussed in the study[c1], with information on activated (✓) or deactivated (–) frazil interaction mechanisms

| Series [c2]ID | $\rho$ | $C_{x,n}$ | $\nu$ | $F_D$ |
|---|---|---|---|---|
| $\mathcal{F}_0$ | – | – | – | – |
| $\mathcal{F}_\rho$ | ✓ | – | – | – |
| $\mathcal{F}_{Cx}$ | – | ✓ | – | – |
| $\mathcal{F}_\nu$ | – | – | ✓ | – |
| $\mathcal{F}_{F_D}$ | – | – | – | ✓ |
| $\mathcal{F}_{all}$ | ✓ | ✓ | ✓ | ✓ |

which is treated as a reference case for the remaining simulations) to $\mathcal{F}_{all}$ in which all four interaction mechanisms described in section 3.3.3 are activated.

The model equations are solved with a time step $\Delta t = 0.25$ s, so that the Courant number is $\ll 1$. Each simulation is 18 h long. The last 12 h of each run (corresponding approximately to one inertial period) are used in the result analysis. [c3]As shown in the examples in Supplementary Note S4, the spin-up time of 6 h is sufficient for a quasi-stationary state to develop, with stable domain-averaged vertical profiles of frazil volume fractions (and other important OML characteristics, not shown).

In the following section, time-averaged and horizontally averaged variables are denoted with brackets, $\langle \cdot \rangle$, and with a bar, respectively. Anomalies from the averaged values are denoted with primes.

[c3] *Text added.*

## 5 Modelling results

### 5.1 Series $\mathcal{F}_0$: no frazil→hydrodynamic coupling

The results of simulations from series $\mathcal{F}_0$, in which the hydrodynamic part of the model is unaffected by the presence of frazil, can be used to analyze the general behavior of the model in the range of forcing conditions considered, as well as to describe the redistribution of frazil particles of different sizes by the OML currents. Series $\mathcal{F}_0$ serves also as a point of reference for subsequent model runs.

Figure 5 shows the vertical profiles of the time- and domain-averaged current velocities $\langle \bar{u} \rangle(z)$, $\langle \bar{v} \rangle(z)$, momentum flux $\langle \overline{u'w'} \rangle$, $\langle \overline{v'w'} \rangle$, and velocity variance $\langle \overline{u'u'} \rangle$, $\langle \overline{v'v'} \rangle$, $\langle \overline{w'w'} \rangle$ for different combinations of $U_a$ and $T_a$. The overall vertical structure of the mean flow is similar to that known from earlier studies and predicted by theory: the $x$ (along-wind) component of velocity exhibits positive values in a thin layer at the surface and negative elsewhere, compensating the positive Stokes drift. The $y$ (across-wind) component is negative (in the Northern Hemisphere) and decreases with depth, the faster the lower the wind speed. Notably, when the air temperature is low and convection is present (continuous lines in Fig. 5), the velocity profiles are close to vertical over most of the OML, with pronounced vertical velocity gradients only within relatively thin layers at the top and bottom of the OML. In the convection-dominated case with $T_a = -20°$C and $U_a = 5$ m/s, the variance of the horizontal and, especially, the vertical velocity components is large over the entire OML due to the dominating influence of convective cells on the 3D flow structure. With weak forcing ($T_a = -1.5°$C and $U_a = 5$ m/s), the flow is mostly limited to the upper parts of the OML. In the remaining cases (yellow and red curves in Fig. 5), the vertical profiles of momentum flux and velocity variance tend to be similar, with very uniform, nearly linear variability far from the top and bottom of the OML.

The average vertical distribution of frazil (Fig. 6a,d,g) is a result of competition between mixing, tending to homogenize frazil volume fractions over the entire OML, and ice buoyancy, tending to concentrate frazil at the sea surface. Not surprisingly, the shape of the equilibrium profiles of $\langle \bar{c}_i \rangle(z)$ changes with the crystal size, and thus floatability. The smallest crystals (Fig. 6a) exhibit nearly uniform profiles under all analyzed conditions except for the lowest wind speed and high air temperature (dashed blue line). The largest crystals (Fig. 6g) tend to accumulate in the surface layer and, especially when convection is weak, their mean concentration exhibits a clear vertical gradient. Under weak wind/weak convection, the area-averaged surface volume fraction $\langle \bar{c}_3 \rangle$ within the top-most model layer exceeds 10 times the OML average; combined with very high variance at the surface, this means that, locally, ice concentration reaches values $> 0.2$, corresponding to grease ice (dashed blue lines in Fig. 6g,h). For the smallest crystals, concentration variance is very small, with maxima in the lowest parts of the OML, where the plumes of frazil-laden OML water interact with the water from the pycnocline [c1](as can be seen in Fig. 5[c2]a,b, in simulations with the strongest wind the horizontal flow reaches to the bottom of the OML, which combined with strong vertical currents in downwelling regions leads to mixing with water underlying the OML). In the case of the largest crystals, which only occasionally reach the bottom of the OML, there is no corresponding deep maximum of $\langle \overline{c_3'c_3'} \rangle$ (Fig. 6h).

---

[c1] *Text added.*

[c2] *Text added.*

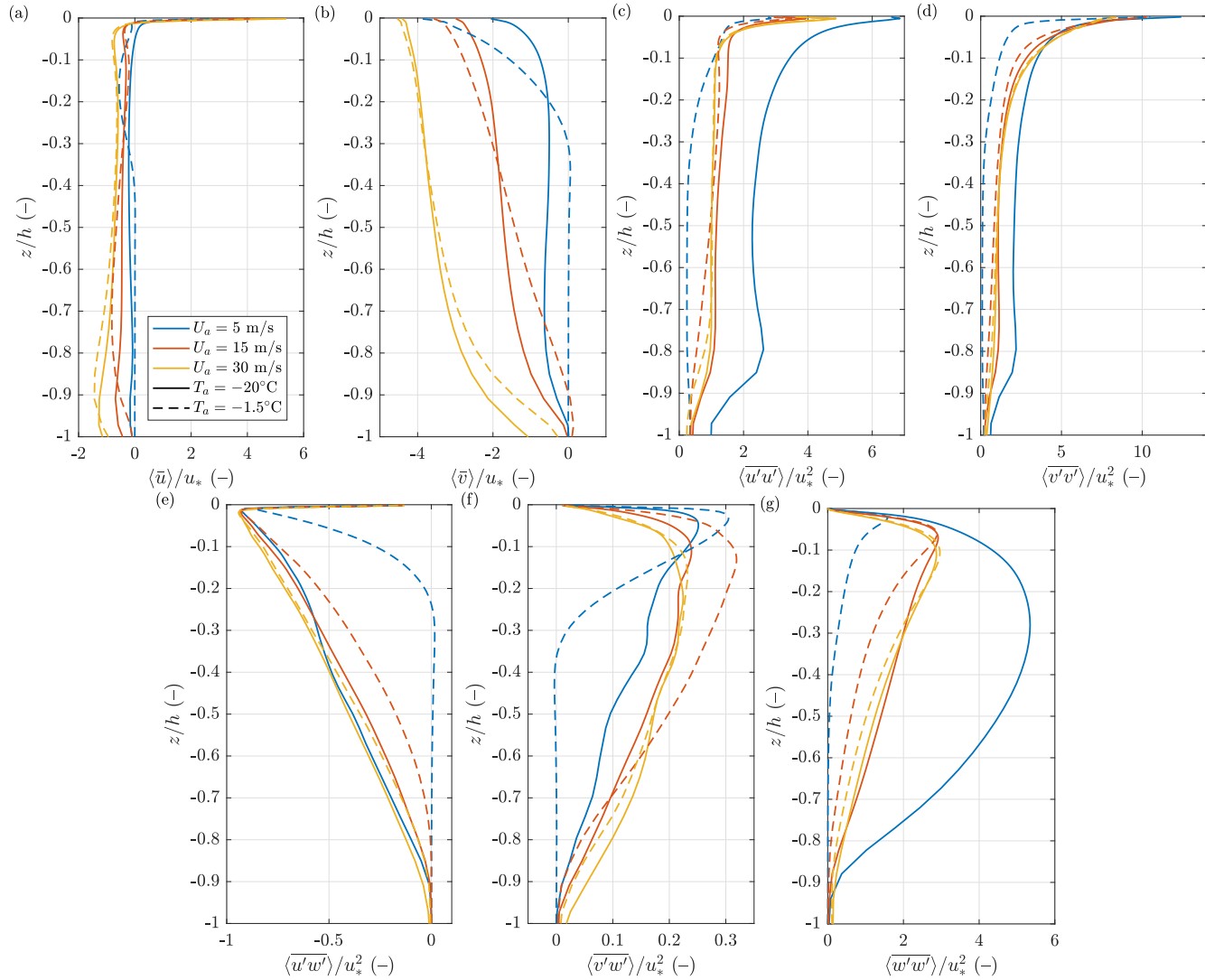

**Figure 5.** Vertical profiles of the mean velocity (a,b), momentum flux (e,f) and velocity variance (c,d,g) for different combinations of $U_a$ (colors) and $T_a$ (line styles). Results of simulations without frazil coupling (series $\mathcal{F}_0$). All values are normalized with the respective friction velocity $u_*$.

The strong near-surface maxima of both $\langle \bar{c}_3 \rangle$ and $\langle c_3' c_3' \rangle$ are manifestations of clustering of large crystals within the upper 1–2 m of the water column, as illustrated for two selected cases in Fig. 7. Under calm conditions, this process is so effective that the volume fraction $c_3$ over a substantial part of the sea surface is close to zero (yellow curve and lower inset in Fig. 7a). Under the most turbulent conditions considered here, even the largest frazil fraction undergoes substantial mixing, but the preferential concentration still takes place and leads to wide probability distributions of $c_3$ at the surface (Fig. 7b).

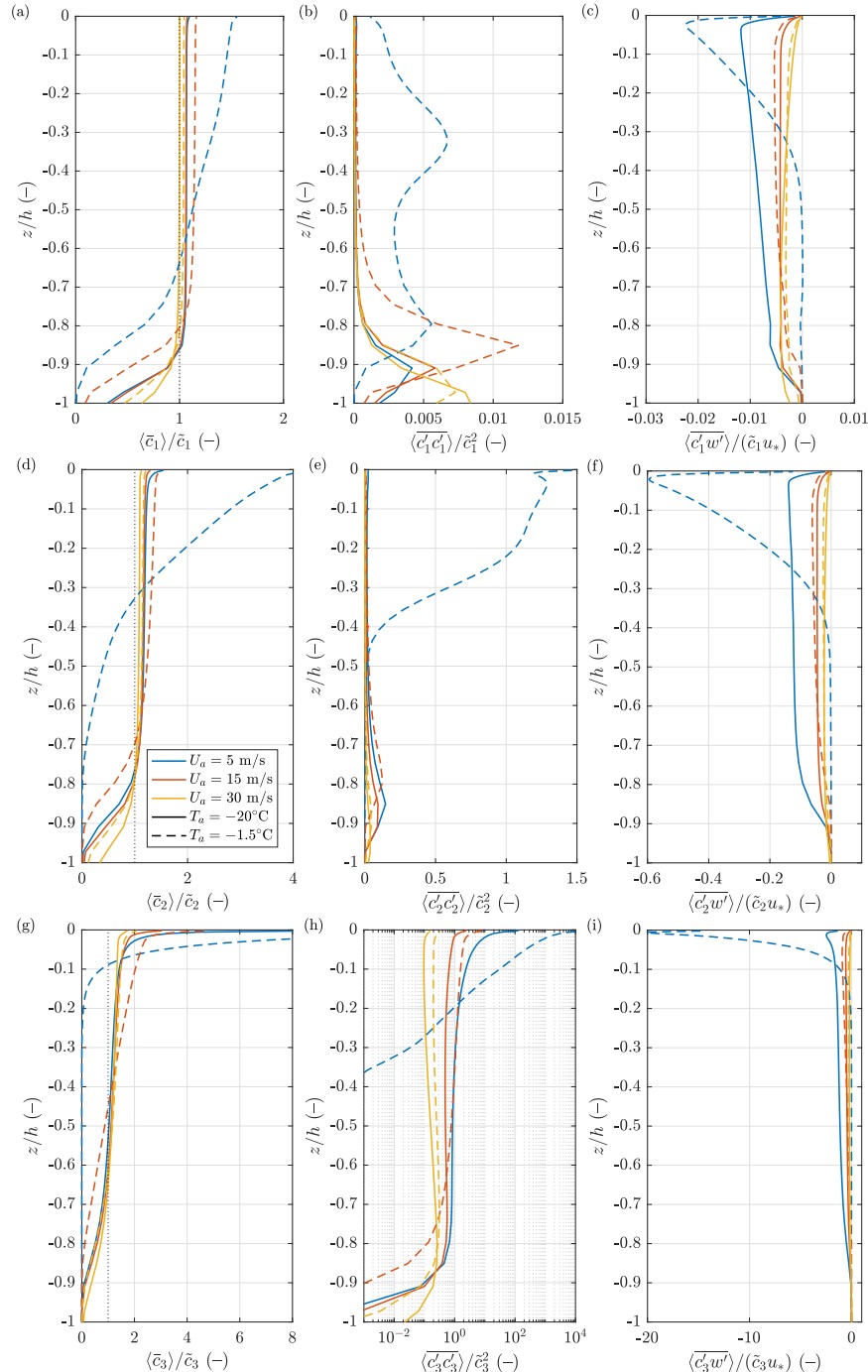

**Figure 6.** Vertical profiles of the mean (a,d,g), variance (b,e,h) and vertical flux (c,f,i) of the frazil volume fraction for size classes 1–3 for different combinations of $U_a$ (colors) and $T_a$ (line styles). Note different $x$ axis ranges in the plots; in (h), the $x$ axis is logarithmic.

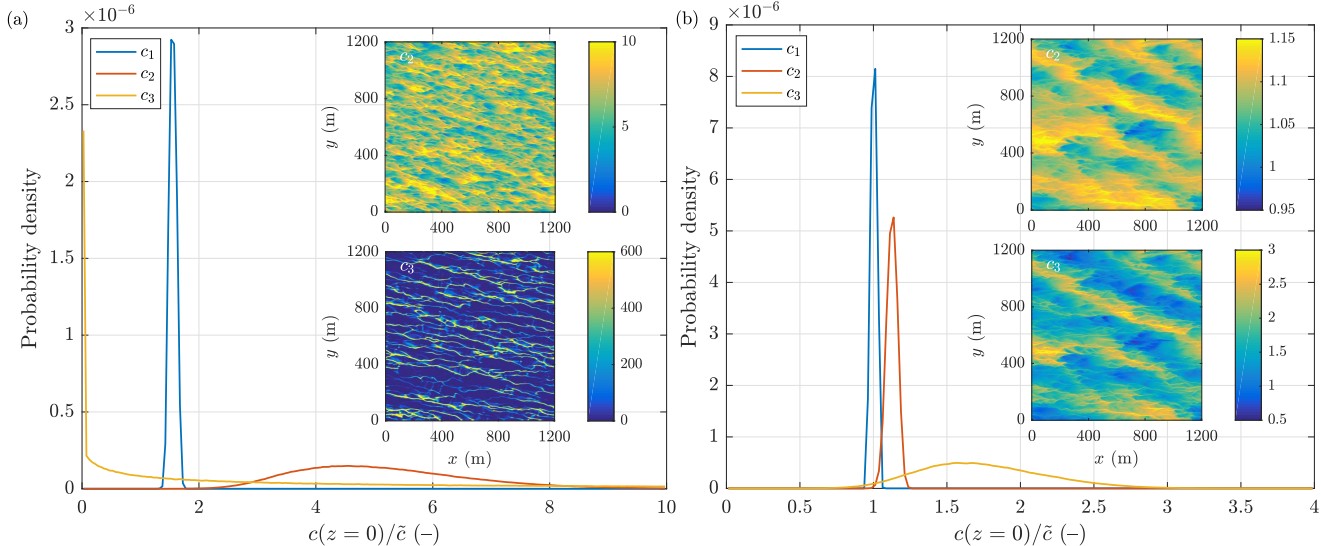

**Figure 7.** Distribution of frazil at the sea surface in simulations with: (a) $U_a = 5$ m/s, $T_a = -1.5°$C, (b) $U_a = 30$ m/s, $T_a = -20°$C. The main plots show probability distributions of frazil volume fraction (normalized with $\tilde{c}$) [c3]at the surface ($z = 0$) for all three size classes. [c4]All curves shown integrate to the value of 1. The insets show snapshots of $c_i/\tilde{c}_i$ for $i = 2$ (top) and $i = 3$ (bottom).

As can be seen in Fig. 6, the values of $\langle \overline{c_i'w'} \rangle$ are negative in all cases analyzed, i.e., positive anomalies of frazil concentration occur in regions with negative vertical velocities, which in turn are located under the surface convergence zones. The zoomed panels in Fig. 8 clearly show that relationship, as well as the fact that the individual convergence zones tend to be narrow, elongated and oriented approximately in the wind/wave propagation direction. The model reproduces also the characteristic pattern of Y-shaped junctions formed by those zones, known from observations and earlier modelling studies on Langmuir turbulence (Thorpe, 2004). Remarkably, the maps of ice concentration at the domain-scale reveal two overlapping patterns, one related to the locally elevated values of $c$ in individual convergence zones, and the other in the form of wider, long streaks oriented obliquely to the wind/wave direction, with typical spacing of hundreds of meters. Accordingly, although it might seem surprising, the correlation between $c$ and horizontal divergence $D$ at the surface is low (in the example illustrated in Fig. 8 it equals $-0.37$). It increases significantly when the local anomaly of $c$, computed over an area $l^2$ surrounding a given location, is used instead of $c$ itself (e.g., $-0.71$ for $l = 18$ m in the example in Fig. 8). The larger-scale streaks of frazil, although clearly discernible in all modelling results with wind $\geq 15$ m/s, have irregular shapes and spacing and their form constantly evolves in time, which, qualitatively, agrees with observations (see, e.g., photographs and satellite images in Eicken and Lange, 1989; Ciappa and Pietranera, 2013; Hollands and Dierking, 2016; De Pace et al., 2019). In particular, it can be seen in Fig. 7 of Ciappa and Pietranera (2013) that each streak has its own sub-structure and consists of several smaller patches, and that the wave direction is not aligned with streak orientation (see also Hollands and Dierking, 2016). The observed separation between frazil streaks of 300–500 m, reported in the cited studies, is comparable with that obtained in our simulations. The example in

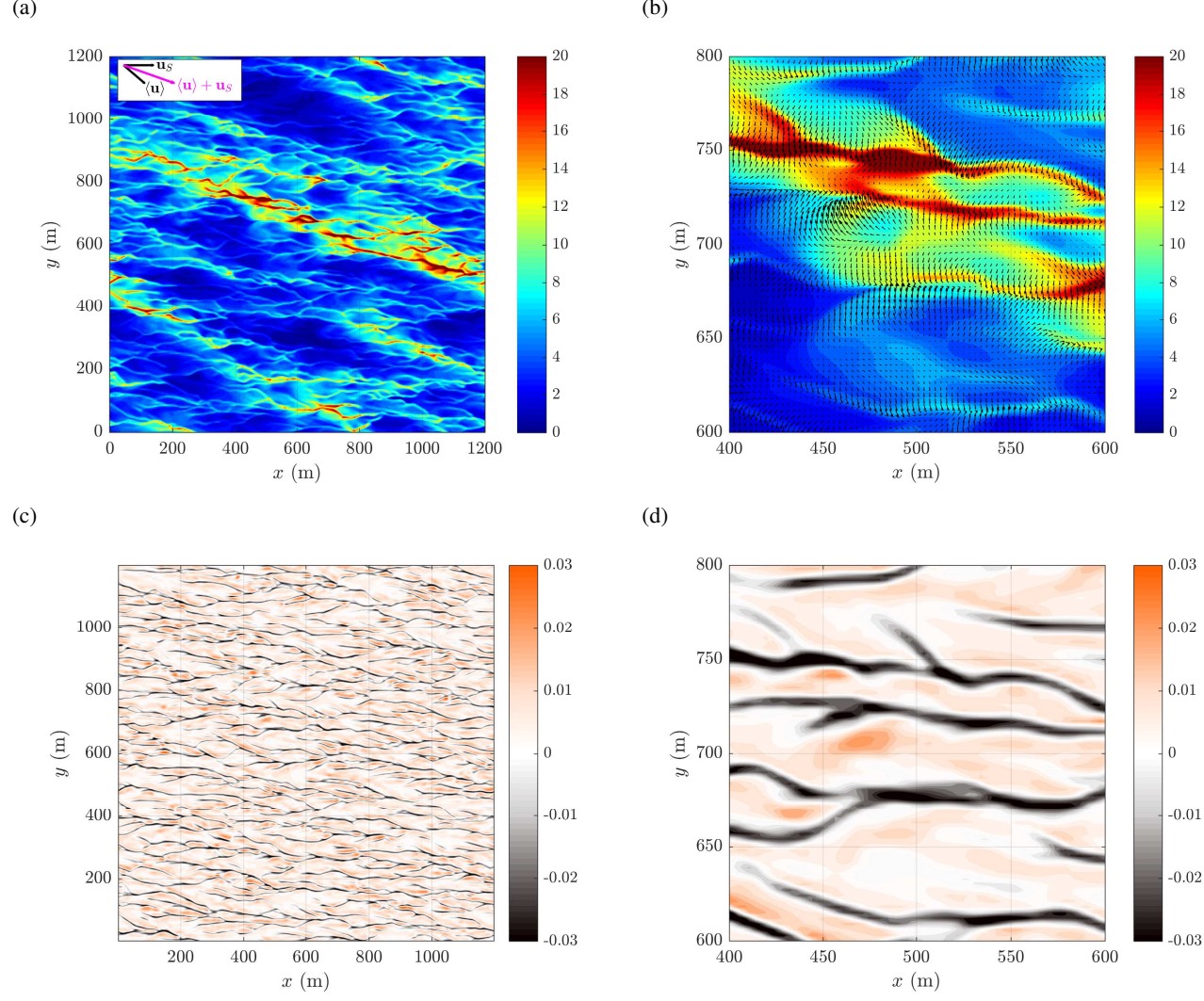

**Figure 8.** Snapshots of the frazil volume fraction $c_3/\tilde{c}_3$ (a,b) and horizontal divergence $D$ (c,d; in s$^{-1}$) at the surface, $z = 0$, from simulations with $U_a = 15$ m/s and $T_a = -1.5°$C. Maps in (b,d) are zoomed fragments of those in (a,c). In (b), arrows show the velocity anomaly $(u - \langle u \rangle, v - \langle v \rangle)$. The inset in (a) shows the vectors of mean velocity $\langle \mathbf{u} \rangle$ and Stokes velocity $\mathbf{u}_S$ at the surface.

Fig. 8, as well as other simulated cases (not shown) suggest that the overall orientation of the streaks corresponds approximately to the net direction of the surface Stokes drift $\mathbf{u}_S$ and the domain-mean surface current $\mathbf{u}$ (see inset in Fig. 8a).

As the lengths of the streaks are comparable with the size of the model domain, which might influence their formation and properties, a test simulation was performed for the case discussed above ($U_a = 15$ m/s, $T_a = -1.5°$C), with all model settings unchanged, but over a larger domain with dimensions $2L_x \times 2L_y$, i.e., 2400$\times$2400 m$^2$. The results are presented in Supplementary Note S5. As can be seen, the overall pattern of the surface ice concentration, as well as the bulk characteristics

of the OML circulation and frazil transport, are insensitive to the domain size (the only noteworthy differences occur for $\langle \overline{c_i' w'} \rangle$, but they never exceed 4–5%). Therefore, although some influence of the limited domain size and periodic boundaries on the streak geometry cannot be ruled out, the general observations made above should remain valid.

[c1]Finally, it is also worth stressing that the vertical distribution of frazil and the basic properties of streaks, described above, are not sensitive to the latitude at which the model domain is located (see Supplementary Note S6 for example results).

## 5.2  Modification of OML circulation by frazil-related processes

As shown in Table 3, the reference simulations from series $\mathcal{F}_0$ were followed by five series in which just one or all four frazil-related processes were activated. Overall, the effects due to the net viscosity and transfer coefficients at the sea surface (series $\mathcal{F}_\nu$ and $\mathcal{F}_{C,x}$) are minor compared to those related to net drag force and[c2], especially, the net density (series $\mathcal{F}_{F_D}$ and $\mathcal{F}_\rho$). [c3]This is particularly true in the case of the mean characteristics of the OML circulation and of frazil distribution are concerned. Consequently, the differences between the results of $\mathcal{F}_0$ and $\mathcal{F}_{\text{all}}$ – which are substantial for most combinations of $T_a$ and $U_a$ – are dominated by the effects related to the lowered density of the ice–water mixture and[c4], to a lesser extent, to the drag force induced by the rising ice crystals on the surrounding water. Figures 9 and 10 show an example for $U_a = 30$ m/s, $T_a = -20°$C, for all six series. The conclusions from this case are valid for the other forcing combinations as well.

As expected, in simulations in series $\mathcal{F}_{\text{all}}$ and $\mathcal{F}_\rho$ the frazil is concentrated closer to the sea surface. Even in the case of the finest size class, although $\langle \overline{c_1' c_1'} \rangle$ is small, i.e., the small crystals are uniformly distributed horizontally, $\bar{c}_1$ is slightly elevated in the surface layer and its values drop quickly towards zero at 70–80 m (Fig. 11b), i.e., even the most energetic forcing considered is not sufficient to mix the finest frazil fraction over the entire OML – contrary to series $\mathcal{F}_0$, in which some (small) amounts of frazil are mixed into the pycnocline layer (Supplementary Fig. [c6]16b). The coarsest fraction (Fig. 11d) is concentrated in the upper portions of the OML and exhibits strong clustering, manifested in elevated values of both $\bar{c}_3$ and $\langle \overline{c_3' c_3'} \rangle$ at the surface (Fig. 10). High ice concentration within those clusters, with typically $c_3 > 10\tilde{c}_3$ (and up to $100\tilde{c}_3$ locally), means that they are also regions of substantially lowered net density. Therefore, it is not surprising that their presence modifies the local circulation in their direct surroundings. Less obviously, [c7]important features of the entire OML are affected, which in turn influence the form of the clusters themselves: the frazil streaks are narrower, with larger ice concentration gradients at their boundaries (see colour contours in Fig. 12 for an example). The flow in the lower parts of the OML, where no coarse frazil is present (below 60–70 m in the case shown in Figs. 11 and 12) is relatively uniform horizontally, directed against the wind/wave direction (note that Fig. 11a and Fig. 9 show the Eulerian component of the total velocity, without the Stokes drift). In the upper layers, several neighboring zones of different flow directions can be seen, with currents towards S–SE (towards SW–S) correlating with areas of high (low) frazil concentrations, respectively. This relationship between frazil streaks and OML flow can be clearly seen

---

[c1] *Text added.*
[c2] *Text added.*
[c3] ~~, especially if~~
[c4] *Text added.*
[c6] ~~12~~
[c7] ~~importnt~~

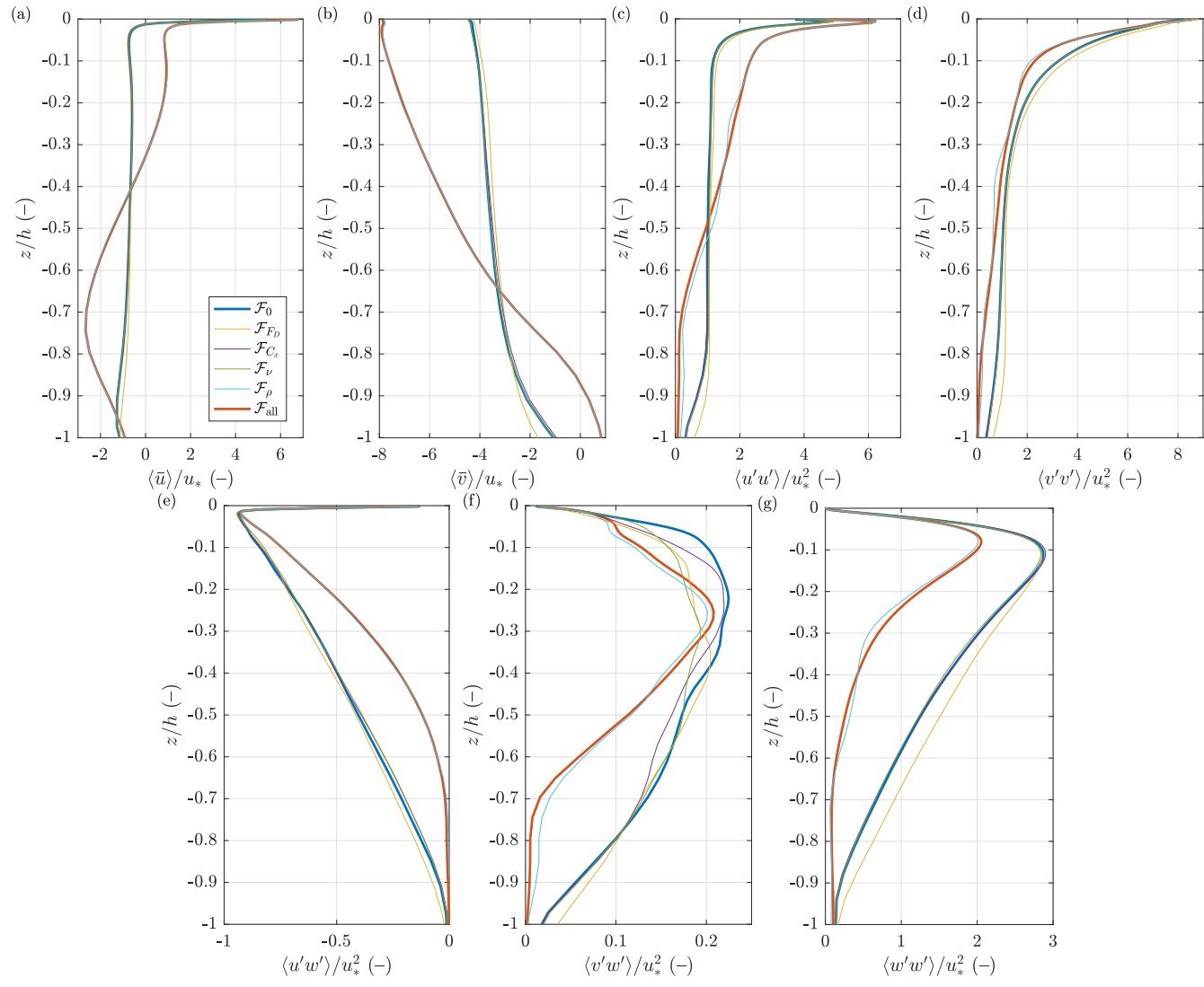

**Figure 9.** Vertical profiles of the mean velocity (a,b), momentum flux (e,f) and velocity variance (c,d,g) for $U_a = 30$ m/s and $T_a = -20°$C, for all six simulation series. All values are normalized with the respective friction velocity $u_*$.

in Fig. 12, in which the anomalies of the vertically integrated velocities are plotted over contours of $c_3/\tilde{c}_3$. Frazil streaks are aligned with zones of anomalies directed to, approximately, ESE, and the strongest flow in the opposite direction takes place in regions with the lowest frazil concentration. An interesting feature are anticyclonic (cyclonic) eddies visible to the right (left) of frazil streaks when looking in the downwind direction, separating from each other areas of approximately linearly oriented flow anomalies. Those eddies are small, with diameters of $\sim$100 m, and are very different from those present in the results of series $\mathcal{F}_0$ (Supplementary Fig. <sup>c8</sup>17), which are larger, dominate the depth-averaged flow, and do not have obvious connections

c8 ~~13~~

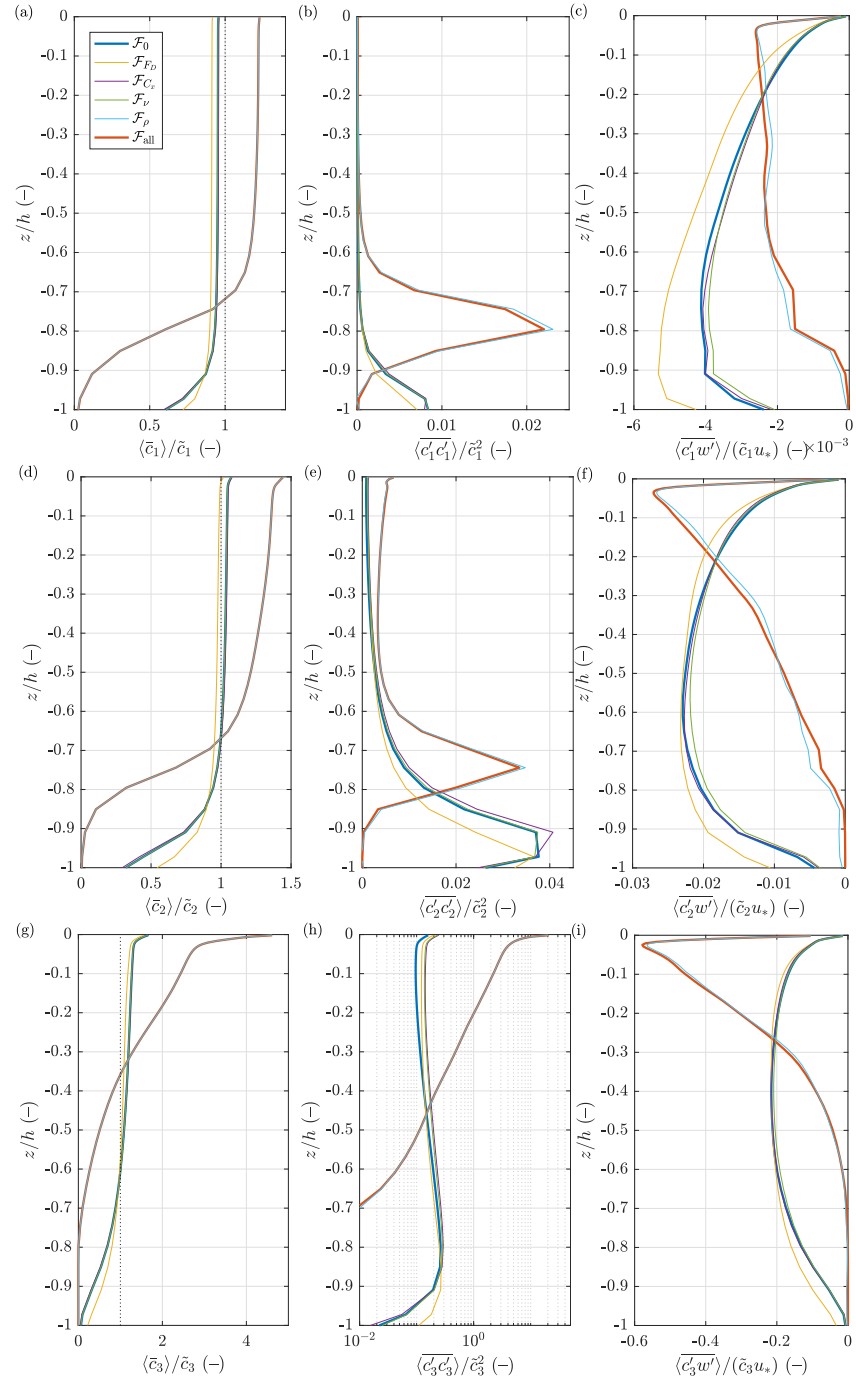

**Figure 10.** Vertical profiles of the mean (a,d,g), variance (b,e,h) and vertical flux (c,f,i) of the frazil volume fraction for size classes 1–3 for $U_a = 30$ m/s and $T_a = -20°C$, for all six simulation series. Note different $x$ axis ranges in the plots; in (h), the $x$ axis is logarithmic.

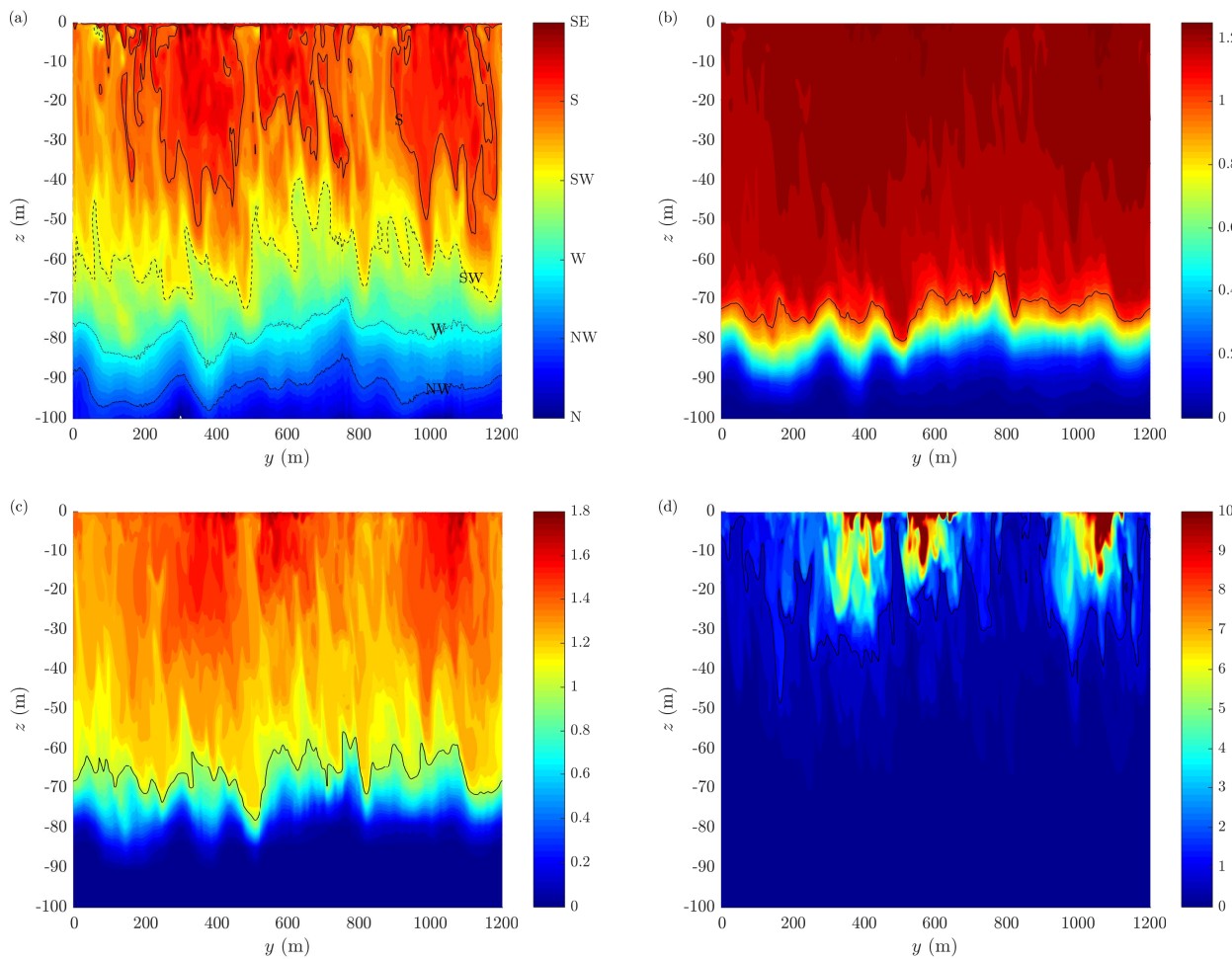

**Figure 11.** Direction of the horizontal flow (a) and frazil volume fraction $c_i/\tilde{c}_i$ (b–d) along a cross-section through the model domain perpendicular to the wind/wave direction ($x = 600$ m in Fig. 12). In (a), a 'to'-convention is used, i.e., S means current flowing *to* the south and so on. In (b)–(d), black contours show the value of 1. Results of simulations with $U_a = 30$ m/s, $T_a = -20°$C, series $\mathcal{F}_{\text{all}}$. See Supplementary Fig. [c5]16 for analogous plots from series $\mathcal{F}_0$.

with frazil concentration – they mostly reflect the convective circulation spanning the entire depth of the OML, manifested also in the cross-sections in Supplementary Fig. [c9]16. Thus, by modifying the net density frazil reduces the vertical range of the convective circulation, disturbs the uniform vertical structure of convective cells, and contributes to the formation of organized, relatively regular depth-averaged flow patterns in the OML.

5    A remarkable difference between the results of series $\mathcal{F}_0$ and $\mathcal{F}_{\text{all}}$ is also related to the vertical water velocities. Without frazil→hydrodynamics coupling, as many studies of OML turbulence and material transport have shown (Chamecki et al.,

---

[c9] 12

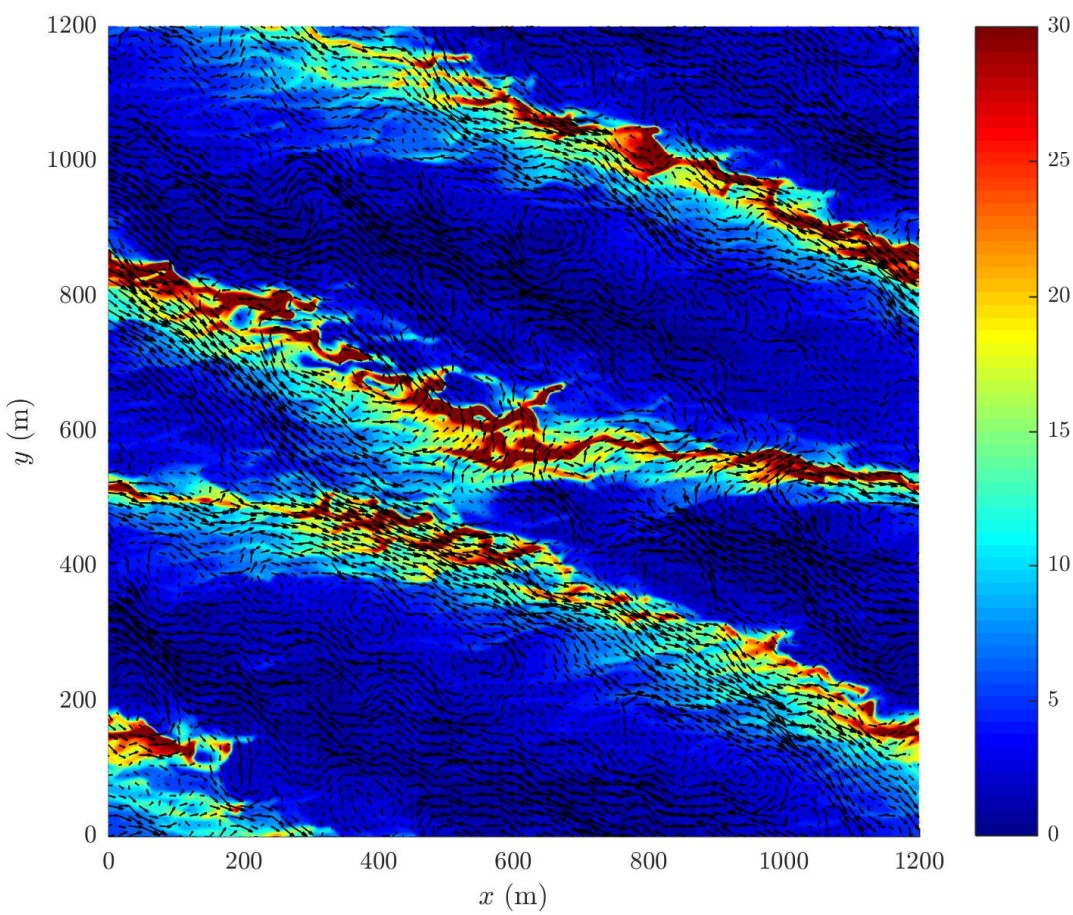

**Figure 12.** Volume fraction of the largest frazil size class $c_3/\tilde{c}_3$ at the sea surface (colour) and anomalies of the vertically integrated horizontal currents (arrows) in simulations with $U_a = 30$ m/s, $T_a = -20°$C, series $\mathcal{F}_{\text{all}}$. For better visibility, only every 5th arrow in each direction has been plotted.

2019)[c1], buoyant particles tend to accumulate in zones over the strongest downward currents, where the surface convergence is strong. This produces correlation between positive anomalies of the particle concentration at the sea surface and variance (or standard deviation) of the vertical velocity component $w$, observed in our series $\mathcal{F}_0$ (Supplementary Fig. [c2]18). In series $\mathcal{F}_{\text{all}}$ the pattern is very different, as shown in Fig. 13: the most vigorous vertical flow occurs in regions between frazil streaks, and

5    within streaks vertical velocities are significantly reduced – although positive anomalies of $c$ still occur over areas of negative

---

[c1] reference added
[c2] ~~14~~

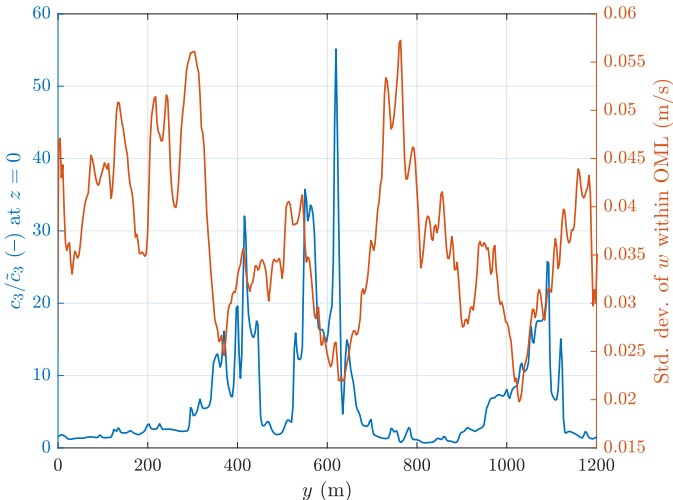

**Figure 13.** Volume fraction of the largest frazil size class, $c_3/\tilde{c}_3$, at the surface (left axis) and standard deviation of the vertical velocity $w$ within the OML (right axis) in the same situation as shown in Figs. 11 and 12 (section along $x = 600$ m).

$w$. In other words, the vertical overturning within streaks is not shut down, but its intensity is reduced. Obviously, this effect might be stronger when the overall ice concentration will be higher than considered here.

When analyzing the influence of individual frazil coupling mechanisms, it is worth noting the role of the ice-induced drag in series $\mathcal{F}_{F_D}$. In many ways it is acting opposite to the density-related effects described above: it leads to vertically more uniform velocity and frazil concentration profiles. Its effects are particularly strong in simulations with lower wind speeds and higher air temperatures, when the results $\mathcal{F}_{F_D}$ exhibit less regular circulation cells (and consequently less well developed frazil streaks) and significantly higher velocity variance $\langle \overline{u'u'} \rangle$, $\langle \overline{v'v'} \rangle$ and, especially, $\langle \overline{w'w'} \rangle$ (not shown).

Finally, coming back to series $\mathcal{F}_{C,x}$ and $\mathcal{F}_\nu$: the vertical profiles of scaled quantities analyzed in Figs. 9 and 10 are very similar to those from series $\mathcal{F}_0$, but the domain-averaged momentum and heat transfer between the ocean and the atmosphere is lowered in $\mathcal{F}_{C,x}$, which leads to generally lower current velocities. The results of series $\mathcal{F}_\nu$ are sensitive to the total amount of ice present in the OML – the influence of frazil on the net viscosity might be significant when $\tilde{c}$ is larger than in the present simulations.

## 6  Summary and discussion

As mentioned in the introduction, formation of frazil streaks on the sea surface was attributed to the Langmuir circulation already in the first publications on that matter (Martin and Kauffman, 1981; Weeks and Ackley, 1982; Eicken and Lange, 1989). It has remained the 'standard' explanation throughout the years, although the arguments supporting it have been of descriptive, superficial nature, qualitative rather than quantitative – the details of the 3D OML circulation in conditions favorable for frazil formation have been outside of the main focus of sea ice research. To the best of our knowledge, no studies have been devoted to

the analysis of size, shapes and other properties of frazil streaks and their dependence on the wind, wave and convective forcing. Somewhat ironically, the fact that the observed streaks have irregular shapes and often join with their neighbors has been sometimes used to suggest that Langmuir circulation alone cannot explain their formation and behavior. Progress in research on OML circulation has shown that exactly those features are distinctive of Langmuir turbulence and that Langmuir cells only

rarely have regular shapes predicted by the classical theory (Thorpe, 2004). The results of our idealized simulations suggest that, at sufficiently high wind speed, the patterns of frazil concentration at the surface emerge from multi-scale processes, with locally elevated ice concentrations in individual surface convergence zones, oriented approximately in the wind/wave direction, and larger-scale, wide frazil bands, oriented obliquely to the wind/wave direction (Figs. 7, 8, 12)[c1]. Although this result seems to qualitatively agree with observations (e.g., the streaks visible in Fig. 1 are too large and too widely separated

to be associated with individual Langmuir cells just a few kilometers offshore, i.e., in strongly fetch-limited situation), more extensive simulations covering a wider range of parameters (in particular, different values of fetch and angle between wind and waves) are necessary to formulate reliable, quantitative hypotheses regarding the spacing and orientation of frazil streaks. Similarly, a different model setting is necessary to analyze the evolution of frazil streaks with increasing distance from the shore, i.e., with spatially-varying wave (and possibly wind) forcing. Certainly, observational data are vital in order to test those

hypotheses and to confine poorly constrained parameters of the models. In the present simulations, several parameters, e.g., the Prandtl number or the coefficient in the Smagorinsky model of horizontal viscosity, have been set arbitrarily, as no data are available that would allow to determine their "proper" values. Our initial model tests suggest, in agreement with what is known from LES OML modelling in general, that the mean vertical profiles of current velocity and frazil concentration are insensitive to the choice of model parameters, especially those describing subgrid-scale mixing, but the velocity/concentration

variance is affected – for example, lowering the Prandtl number leads to slightly stronger clustering of frazil at the surface. Thus, constraining the unknown coefficients is important for realistic representation of preferential concentration of buoyant material in the OML.

[c1]Even in the idealized model setup used in this study, the number of different combinations of model settings – forcing conditions, coefficients in various (semi-)empirical formulae describing physical processes, alternative mathematical formu-

lations of those processes, etc. – is very large. Therefore, confronted with extremely high computational costs of the model, we were only able to explore a small subspace of the whole, multi-dimensional model parameter space. As described in sections 3 and 4[c2], initial sensitivity studies were used to assess the role of some, selected settings. However, several parameters have been set arbitrarily and without further, more extensive simulations, supported with observational data, care should be taken by extending the results presented in this paper to conditions outside of the range of those considered here. Also, some

formulations used in the new frazil module are well established (or even rather obvious, as is the case with the net density of ice–water mixture computed from formula (18)[c3]), while other are based on solutions used in general-purpose mixture models and their suitability for frazil in the OML remains to be assessed. Moreover, it should be remembered that some of the settings

---

[c1] reference to figures added

[c1] *Text added.*

[c2] *Text added.*

[c3] *Text added.*

used in this paper are not suitable for simulations with frazil thermodynamics activated. This is certainly true for the number of frazil size classes $N_f$. Whereas the very small number of classes used here turned out sufficient for capturing the processes of interest (initial tests with $N_f = 10$ led to the same conclusions as those with $N_f = 3$), large number of size classes and a wide range of sizes covered are absolutely necessary to properly simulate the evolution of the crystal size distribution related to thermodynamic processes.

Another, serious limitation of the model used in this study[c1], mentioned at the end of section 3.3.3, is related to the formulation of the governing equations, which are valid only for low ice volume fractions. In the simulations discussed above care has been taken that this assumption has not been violated, but, evidently, this narrows the range of applicability of the model. In the earlier models by Holland and Feltham (2005), Heorton et al. (2017) or De Santi and Olla (2017), the equations are formulated for low ice concentrations as well, but it is assumed that the frazil reaching the sea surface precipitates out of the water column, forming a layer of grease ice at the top of the OML. Moreover, several simplifying assumptions regarding the properties of that layer are made, e.g., it has a constant ice volume fraction and once ice enters that layer, it cannot re-enter the water below, i.e., grease ice is in many ways "decoupled" from the rest of the OML. Crucially, no equations describing the motion of that layer are formulated. This approach is acceptable in one-dimensional settings analyzed in those papers, but cannot be transferred to a 3D model setup, in which the transient dynamics of a (discontinuous) grease layer and its interactions with the surrounding water have to be taken into account. Thus, multiphase mixture models, valid for an arbitrary number of frazil classes and for $0 \leq c_i \leq 1$, seem the most appropriate direction in developing future OML–frazil models. Although several elements of the frazil module in our model, including the method of calculating the ice-induced drag force or the net viscosity, are based on methods used in mixture models, it is just a first step towards a proper treatment of the processes and coupling mechanisms involved. It is also worth stressing in this context that some of the effects analyzed here, which turned out minor within the range of ice concentrations considered in this study – like, for example, the influence of frazil on the net viscosity – might become relevant at higher ice concentrations.

A very interesting aspect of the results presented in this paper is the "sorting", or de-mixing of initially uniform mixture of ice crystals of different sizes due to their different floatability. Our results agree with those by Clark and Doering (2009), who observed in a laboratory that small particles were more uniformly distributed in the water column than large ones, which tended to exhibit concentration maximum at the surface, especially under low-turbulence conditions. That inherent sorting mechanism might produce several interesting effects when combined with processes not included in the present model version – secondary nucleation, [c2]flocculation, crystal growth and melting, and other thermodynamic processes which are affected by the size distribution of crystals, and consequently might occur at different rates in surface divergence zones, where only the smallest crystals are present and the total ice volume fraction is relatively low, and in convergence zones, in which the largest crystals accumulate and the width of the crystal size distribution is very wide. It is a subject for subsequent studies to investigate how those 3D dynamic–thermodynamic interactions influence the total frazil production rates and other bulk OML properties, and thus whether they should be parameterized in larger-scale models. The same is true for size-dependent

---

[c1] *Text added.*

[c2] *Text added.*

transport of frazil within the OML. As described in the previous section, in coupled simulations (series $\mathcal{F}_{\text{all}}$) frazil streaks form in zones of positive velocity anomalies, coarser frazil fractions contribute the dominating part to the total ice concentration within those streaks, and due to their higher floatability coarse fractions accumulate closer to the surface. Thus, apart from stronger Eulerian currents within streaks, larger ice crystals experience also stronger Stokes drift, and in consequence are

5 transported faster – and in a different direction – than small crystals, well mixed over the entire OML. This illustrates how the local frazil–hydrodynamics coupling might modify both the bulk behavior of the OML and frazil evolution during frazil/grease ice formation.

*Code and data availability.* The code of the CROCO model is freely available at http://www.croco-ocean.org/download/croco-project/. The extended code necessary to reproduce the simulations presented in this paper, together with input scripts and modelling results, can be

10 obtained from the corresponding author.

*Author contributions.* All authors contributed to planning of the research and to discussion and analysis of the results. A.H. developed the numerical model, performed the simulations and wrote the text.

*Competing interests.* The authors declare no competing interests

*Acknowledgements.* This work has been financed by Polish National Science Centre project no. 2018/31/B/ST10/00195 ("Observations and

15 modeling of sea ice interactions with the atmospheric and oceanic boundary layers"). All calculations were carried out at the Academic Computer Centre (TASK) in Gdańsk, Poland. The code of the model is based on the Coastal and Regional Ocean COmmunity Model (CROCO), provided by http://www.croco-ocean.org.

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
