# Peer review of "High-resolution simulations of interactions between surface ocean dynamics and frazil ice"

_The Cryosphere, 2020_

## Referee Comment (RC1) · Anonymous Referee #1 · 10 Aug 2020

**1   Summary**

This paper investigates the dynamics of the ocean mixed layer (OML) in the presence of frazil and grease ice using Large Eddy Simulation (LES). It studies the effect of wind-driven, convective and Langmuir turbulence on frazil ice, and also the effect of frazil ice on OML dynamics. The turbulent flow leads to segregation of the frazil ice and the formation of streaks of ice on the surface, which are qualitatively similar to field observations. The frazil ice can strongly influence OML dynamics, primarily through its effect on buoyancy. The study suggests several avenues for future research.

I think that the topic of the study is interesting and novel in several aspects. Previous observations have only been interpreted in a qualitative fashion and previous models have been one-dimensional rather than the three-dimensional calculations presented here. The paper is very well written and the analysis performed is thorough with most of the limitations clearly explained. There are a few relatively small weaknesses discussed below which the authors can use to revise their manuscript. However, overall, I think the paper is excellent and should be *accepted subject to minor revisions*.

**2   General comments**

1. **Model formulation:** there are some limitations/assumptions of the model that should be discussed more clearly or considered in further or future calculations.

   The hydrodynamic equations (1–4) assume that the concentration of frazil is small. If this were relaxed, they would need terms like $(1-C)$, where $C$ is the total frazil concentration, in various places (see e.g. Jenkins and Bombosch, 1995).

   The frazil model doesn't consider crystal growth (which is a reasonable starting point and is well discussed). However, I didn't understand why only three crystal sizes were used rather than a much better-resolved crystal size distribution? Presumably, this is not a very expensive part of the overall calculation? Was the sensitivity to the number of crystal size classes tested? It will certainly be essential to include many more when crystal growth and nucleation are considered (as mentioned some of the cited references). Another subtle issue is the assumption that the crystals have a constant aspect ratio. An alternative is to assume they have a constant thickness, which is arguably more reasonable from a crystal growth point-of-view. The crystals remain disk-shaped because it is energetically much easier to grow radially than in thickness.

   The results presented here are clearly very sensitive to the frazil terminal velocity (figure 3). I think the authors should consider comparing their calculations with laboratory data (e.g. of McFarlane et al. 2014). They should also consider crystal-shape effects (assuming eq. 16 wasn't designed for disk-shaped particles).

2. **Sensitivity of results:** The authors choose a particular OML-average volume fraction of 0.00168 for each category, so in total 0.005 (i.e. 0.5%). This is actually rather high. I think there should be better discussion of the sensitivity of results to this choice (e.g. $F_\rho$ must increase with increasing ice concentration, but is the sensitivity linear or are there nonlinear feedbacks?)

3. **Comparison with observations:** The paper makes some comparison with observations, particularly the streaks of ice visible at the surface. However, the comparison is mostly qualitative. This is fairly well discussed in the final section; a forward link could be added in the final paragraph of page 20.

   A more quantitative comparison would be preferable. A starting point would be to devise and calculate statistical measures of the band size and spacing in the numerical calculations and then consider whether these are affected, for example, by wind speed. This could additionally be used to compare plots in Supplementary Fig. 9 quantitatively.

**3 Specific comments and technical corrections**

4. **P2, L20:** 'does have influence' → 'influences'.

5. **P3, L6:** suggest adding review article Daly: Frazil ice dynamics, CRREL Monograph, 84, 46 pp., 1984.

6. **P4, L14:** suggest expanding discussion of laboratory observations.
7. **P5, L3:** explain briefly why turbulent conditions are necessary.

8. **P5, L4:** parenthetical remark a bit confusing, I would delete whole remark and instead change 'buoyancy' to 'convection' or 'buoyancy-driven convection'.

9. **P6, L10–12:** is this good for frazil, especially the bigger crystals?

10. **P12, L9:** should 'd' etc be italicized?

11. **Sec. 3.3.3:** I think this section could have had more discussion of uncertainty. I would imagine that (18) is a more robust relationship than the others.

12. **P15, L9:** where does the latitude come into the calculation? I assume only in Coriolis term but the role of rotation didn't seem to be discussed much.

13. **P15, L12:** vertical boundary conditions on frazil concentration (I saw some earlier discussion of boundary conditions for CROCO in general, but presumably these references don't say anything about frazil).

14. **P17, L1–4:** how/why were these chosen? If you turn on crystal growth in future, results will be extremely sensitive to supercooling.

15. **P17, L16:** I would make it clearer that the phrase 'this choice' is referring only to the uniform distribution, not to all the other choices.

16. **Fig. 5:** Quite busy but just about readable, consider removing intermediate $U_a$.

17. **P20, L12:** expand discussion of interaction with pycnocline

18. **Fig. 7:** I found the main plots confusing and think they need a clearer $x$-label and caption. Is this a horizontal average? Do the PDFs integrate to 1?

19. **Sec. 5.2:** This section gives an impression that buoyancy and drag are similarly important, but the graphs suggest that the all-forcing result is very similar to buoyancy, which suggests buoyancy is much more important than drag.

20. **Fig. 9:** Thin lines very hard to see and distinguish. I would make all lines thicker and use line style to distinguish.

21. **P24. L1:** Typo? (Fig. 11g)?

22. **P24. L5:** Typo in word 'important'.

23. **P28. L19:** In a different way to $F_\rho$?

24. **P30. L17:** Could also mention flocculation?

25. **P30. L31:** The editor may wish to consider the journal's policy about code availability. My opinion is that code by correspondence is less good (in terms of reproducibility) than code made publicly available with a doi.

26. **Supp. eq. (2):** $r$ appears on both LHS and RHS.

---

## Referee Comment (RC2) · Harry Heorton (Referee) · 12 Aug 2020

**High-resolution simulations of interactions between surface ocean dynamics and frazil ice**

Review H. Heorton

This paper documents the development and simulations from a 3D model of the Ocean Mixed Layer OML, within a polynya. The paper is very well presented and I particularly appreciated the extended description of the developments of OML models that allowed for this study to be undertaken.

The paper includes an extended section describing existing methods of analysing the mixing regimes for the OML, and then applying them to an OML with suspended particles. This is then expanded for latent heat polynyas, with the likelihood of each mixing regime discussed. The authors conclude that all three mixing regimes are likely in coastal polynyas but Langmuir turbulence is likely to dominate.

Then follows a full description the model that is thorough and easy to follow for such a complex model.

Two groups of simulations are presented. First a model with no frazil to hydrodynamic coupling under various atmospheric forcing. The mean states of these model runs are described showing the structure of the ocean currents and distribution of frazil crystals. The surface distribution of crystals and lateral currents are presented and contrasted with observations. Then a second group of simulations are presented that show the effect of adding in frazil-related processes. The results from these runs are compared with the first group to show the influence of each process.

The main finding presented are:

Adding the influence of frazil crystals to the net density of the ocean has the greatest influence on the vertical profiles of ice crystals.

The observed surface ice crystal collection is dominated by the largest size class of crystal. These observed surface features, whilst mainly driven by Langmuir circulation are also the result of multi-scale processes.

The results show that ice crystals or different sizes will be 'sorted' by their floatibility, with larger crystals at the surface and smaller crystals at depth. This true in previous work and is also true for this complex 3d modelling study.

I recommend the paper for publication with first a few minor corrections as listed below (mainly relating to referencing within the paper and adding citations). Also I have a few more questions that occurred to me when reviewing the paper that I'd like to see addressed.

Can the authors comment on the chosen initial distribution of frazil ice crystals? I was expecting to see a sensitivity study of this chosen distribution and quantity of suspended crystals. I understand that this study is focussing on the interactions between crystals and the hydrodynamics with no thermodynamics implemented. Did the authors test other initial conditions? I think the paper needs a statement/discussion on the validity of the chosen distribution to allow the reader to understand the context of the simulations and to allow for the interpretation of these results amongst observations of frazil crystals within polynyas and other model simulations. For example in our

paper Heorton (2017) we have many examples of the crystal concentrations at depth, although this is a 1D thermodynamic process model.

Can the authors confirm whether they focus on latent heat or wind driven polynyas? The temperature conditions suggest a latent heat polynya where the temperature of the ocean restricts the growth of ice. But for these conditions what is the expected frazil crystal desnsity and how physically realistic are the choice of crystal concentration and ocean temperature? I realise that these quantities are of secondary importance when thermodynamics are ignore and you focus on the crystal/flud interactions. However an idea of context will help future work where others may wish to compare your results to observations or other models.

I am also interested in the limitations of the model in terms of the intial conditions and time span. I will give some context to my question: We found in our paper that steady state condtions were possible for a wind opened polynya when modelling the thermodynamics properties of ice crystal growth (I am aware that you model a latent heat polynya). A crucial part of the steady state was secondary nucleation of small crystasls. This is crucial as small crystals have lower 'floatability' (as the authors here call it, a useful term) so can be more easily mixed downward, and thus replace the larger crystals that reached the surface (and were removed from the simulation in our model). This steady state was able to persist for several days of model simulation for particular model parameters and atmospheric forcing. However when chosing other parameters (see the results table in Heorton (2017), such a state was not achievable in the model. For example when reducing the OML turbulent mixing rates it was possible to precipitate all the frazil crytals to the surface and thus break the model. Balancing such phenomena was a major challenge of our model development and therefore I ask whether similar limitation were present in the authors model which contains very sophisticated modelling of the ocean turbulence and small scale flows. In particular, are there cases where all crystals collected at the surface? which model run has the greatest time variation in the vertical distribution of ice crystals? What are the model limitations due to the lack of thermodynamics? As the OML supercooling and secondary nucleation driven frazil crytal 'blooms' occur on time scales shorter than the model simulations shown here, how to the authors think that such processes will integrate with their model. I considered asking you to include time series plots alongside figures (5,6,9,10) to show how much variation occured for each simulation, though I don't think this is neccessary as the rest of the paper is so well presented and very extensive. However I think it a good idea to indicate how much variation from the initial conditions there are, for example how long did the surface increase in c3 for Ua=30 m/s Ta = -1.5 deg case in figure 6 (g,h,i) take to form and how stable was the feature? Did c3 continue to rise throughout the model run (with the layer of increased c3 getting increasingly shallow with time) or is the feature in the figures semi-stable? Similarly with figures (7,8,12), at which time point did such features start to occur and did they then remain for the rest of the model run? When running our model of frazil crystals in the OML, two runs with differing spatial distributions of crystal concentration, typically had differing patterns of time evolution. Do the authors find the same with their model?

Minor points

Pg1
L22 The use of 'obviously' is not very helpful in section with no references.

Pg2
L1-2 I agree with this sentence but it needs references for the situations listed. For example - Air-Ice-Ocean Interaction McPhee 2006

L3-5 again this sentence is good, but I'd like to see some references.
L5 'distinctive feature' definitely needs a reference

L11 and L 16 I think here you refer to sea-ice-climate models, or the sea ice component of a climate model

L 17 a reference to these observations is needed here where they are introduced

L 20 again a reference for the additional physical processes that are caused by the presence of frazil ice are needed.

L 28 I see you have the references included here. A link from the above paragraph to the area of the paper with more detail is needed to aid a reader seeking greater detail.

Pg4

L 19 - 21 consider splitting this sentence up as it is difficult to read. Also this sentence describes the limit of the presented modelling study. Crystal thermodynamics are not implemented. This needs to be very clearly stated.

Pg5

L5 'an earlier idea by' to be removed.

L 12 can you add a definition for the Langmuir stability length

L 26 whilst you have a description of Langmuir turbulence above, a definition of Langmuir circulation and transport will be helpful here.

L 30 I see here you have linked the Langmuir circulation and turbulence. As this section contains many different terms, It would aid the reader to have definition and physics behind Langmuir circulation/turbulence/transport defined at the beginning.

Pg 8

L 18 Does this description of low wind conditions over polynyas come from an observation or your results? Can you describe or give a reference.

Pg 9

L7 Can you expand on 'preferential concentration of frazil within the turbulent OML should be limited to the largest crystals ' please? Do you mean that due to the increased rising velocity of larger ice crystals one only needs consider the largest crystals? If so I find this alarming as previous results show that a there are vastly greater numbers of small crystals for certain cases, particulalry at depth.

Equation 1-4, as the model description is complex, I would like you to briefly describe what physical quantity is being conserved in each of the governing equations. This will greatly aid the readability of the following sections.

Pg 11

L 25, which equation describes the horizontal mixing?

Pg12

Equation 12 in which equations are these transfer coefficients implemented?

Pg 13

L5 can you provide a reference for the Schiller Naumann model and use for frazil crystals.

Pg 14

L 6 what observations?

Pg 15

Figure 4 In my experience frazil volume fractions rarely exceed 0.25, this is what is expected for grease ice, and within the OML will be lower. What is the application of the parameterisation in this figure where a volume fraction of 0 to 0.5 is presented?

Pg 16

L 13 I see the choice of frazil crystal radius is sensible when considering the terminal velocities. However how does the choice of crystal classes and initial concentrations compare to previous studies? A resultant ice thickness of all suspended ice crystals of 0.5 m seems to be a large volume

of ice to be suspended within the OML. Our work with frazil crystal modelling with thermodynamics suggests values or around 0.1 m (Heorton 2017). Also with climate models the initial stages of ice growth typically result in 0.05 m of new ice. Is your chosen crystal distribution specifically for a latent heat polynya? I realise that your model contains no thermodynamics so you need to seed a high amount of crystals.

L 17 did you vary the frazil ice concentration at all?

Pg 18
Table 2 Are all parameters other than Ua and Ta derived from these choices and other model parameters? If so can you say so in the caption.

Table 3 Can you expand on the what the F symbols on the left of the table indicate within the caption? If you're including this table to inform the reader about the what was included in the runs, please make it easy to read!

L 7 is the wind aligned with the x or y axis?

Pg 20
L5 I doubt there are any equilibrium profiles of ice crystals. Can you comment on this, what are the time variations in the mean contours? Are the variation plots spatial or time variation? For your model with no thermodynamics I expect all the frazil ice to eventually be driven by buoyancy and collect at the surface.

L7 are you plots in figure 6 time averaged over 18 hours? Over this time scale I would expect a large change in vertical concentration. However my experience is from a thermodynamic model focusing on supercooling and ice crystal growth. Can you comment on the variance in time of the spatial coverage of ice crystals for the cases presented in figure 6?

L 31 as these patterns continually evolve in time, do the mean (over x and y directions) vertical profiles of crystal concentration and ocean velocity also vary in time? how long do the spatial patterns take to form?

Pg 21
Fig 7 what time point were the snap shots taken? How representative are these snap shots of the model run? How do they compare to initial model conditions? How long do they take to form?

Pg 22
Fig 8, similar comments to fig 7. These two figures are well presented and look great, but I find myself asking how representative are they of the whole model run. Additional information about the temporal changes observed in the results will help.

Pg 28
L2 which studies have shown this?

Pg 29
L9 is this statement referencable?
L14 can you link back to the the figure or discussion where this is presented?
L 34 is your discussions of high and low ice concentrations here relating to the difference between ocean with suspended crystals and a grease ice layer, or are you referring to the total ice concentration within the OML?

---

## Author Comment (AC1) · 17 Sep 2020

**Response to the comments of the reviewers**

We are very grateful for both, very positive reviews of our study and for all the comments and suggestions. They are very valuable not only from the point of view of the present manuscript, but also for our further work on frazil modeling and we will do our best to take them into account.

Before we proceed to answering the individual comments and questions, we would like to comment on those issues that were mentioned by both reviewers.
Although the reviews focus on slightly different aspects of our work and point out its different shortcomings, they are similar in that both suggest rather minor modifications to the present version of the manuscript, but at the same time contain a long list of possible improvements, further research directions and open questions worth answering. We really appreciate all those suggestions – as well as the fact that both reviewers understand that many questions will remain unanswered in the present paper, even in its revised version.
At present we are working on:
1. simulations with thermodynamics, analogous to those presented in the manuscript (i.e., based on an idealized model setup)
2. quantitative analysis of the properties of frazil streaks under different forcing conditions (case studies based on satellite images; statistical measures characterizing the properties of frazil streaks in observations and in our modeling results).

In both cases we do have preliminary results, which will be referred to in some of our answers to the individual comments of the reviewers below, but it is too early to include those results in a publication. Moreover, the manuscript is already quite long, and adding additional issues – none of which can be easily squeezed into a single paragraph – would make it even longer.

In particular, we are aware that we haven't fully explored the multi-dimensional parameter space of our model. There are parameters, assumptions, forcing conditions etc. that have been set rather arbitrarily. We take care that those things are stated clearly in the revised manuscript. In some cases, when initial tests have showed that a given model setting has a limited influence on the model behavior (*in terms of quantities analyzed in our study*), we decided to keep it constant to reduce the number of model runs. The model is extremely expensive computationally, mainly due to its non-hydrostatic part. Even if it is run in parallel on a powerful cluster, the model run time in the present configuration is close to the real time (which might at least partially explain why no similar model analyses have been done so far).

Another issue raised by both reviewers is the small number of crystal size classes that we use in our simulations. The main reason for that, just mentioned, is computational efficiency. Increasing the number of size classes leads to substantially longer computation times. In our initial tests we used 10 size classes (which still is a rather small number), but we observed that the results differed only slightly from those obtained with only 3 classes, and we decided to use fixed $N_f$ = 3 in all simulations described in the paper. We are convinced that none of the general conclusions of our paper are affected by that choice, although of course there might be some quantitative differences in the results of individual model runs. As the reviewers correctly remark, the situation is very different in simulations with thermodynamics – in that case the modeling results, including important bulk quantities like e.g. the total frazil mass, are very sensitive to $N_f$ and to the range of crystal sizes covered. This is known from earlier, one-dimensional studies (including those mentioned by the reviewers) and we were not surprised to observe that in our results as well. We do agree that this issue should be stated more clearly in the text, and we elaborate on that in the revised manuscript. The role of $N_f$ will be analyzed in out subsequent paper presenting results with thermodynamics.

**Response to the comments of Reviewer #1**

General comments

1. Model formulation: there are some limitations/assumptions of the model that should be discussed more clearly or considered in further or future calculations.
The hydrodynamic equations (1–4) assume that the concentration of frazil is small. If this were relaxed, they would need terms like (1-C), where C is the total frazil concentration, in various places (see e.g. Jenkins and Bombosch, 1995).

Yes, and we take care that this fact is stated clearly in the revised manuscript.
In this work, we concentrate mainly on very turbulent conditions in which strong mixing prevents very high ice concentrations at the surface, but obviously more general form of the model equations, with the (1-$C$)-terms included, is necessary if the model is to be applied in a wider range of conditions.

The frazil model doesn't consider crystal growth (which is a reasonable starting point and is well discussed). However, I didn't understand why only three crystal sizes were used rather than a much better-resolved crystal size distribution? Presumably, this is not a very expensive part of the overall calculation? Was the sensitivity to the number of crystal size classes tested? It will certainly be essential to include many more when crystal growth and nucleation are considered (as mentioned some of the cited references). Another subtle issue is the assumption that the crystals have a constant aspect ratio. An alternative is to assume they have a constant thickness, which is arguably more reasonable from a crystal growth point-of-view. The crystals remain disk-shaped because it is energetically much easier to grow radially than in thickness.

As for the number of size classes – see our comment on the first page.
The assumption that the crystals have a constant aspect ratio rather than thickness has been taken from Holland & Feltham (2005), who in turn adopted it from Smedsrud & Jenkins (2004). We admit that we did not investigate how this assumption influences the results. Again, we do not expect that keeping the crystal thickness constant instead of aspect ratio would in any way change the conclusions from our results. But we will certainly analyze the role of crystal shape in our new simulations with thermodynamics, as several aspects of thermodynamic processes are sensitive to that parameter. Thank you for pointing out this issue.

The results presented here are clearly very sensitive to the frazil terminal velocity (figure 3). I think the authors should consider comparing their calculations with laboratory data (e.g. of McFarlane et al. 2014). They should also consider crystal-shape effects (assuming eq. 16 wasn't designed for disk-shaped particles).

Yes, it is true that we should reference some observational data regarding the terminal velocities used in our study. We do that in the revised manuscript.
The values of terminal velocities that we use in the simulations have been selected based on the literature. For instance, McFarlane et al (2014) report rising velocities between $4 \cdot 10^{-4} – 1.3 \cdot 10^{-2}$ m/s, which correspond very well with our values of $10^{-4}$, $10^{-3}$, $10^{-2}$ m/s. Our assumed relationship between the crystal size and terminal velocity (the blue curve in Fig.3) also agrees well with McFarlane et al (2014).
But of course, we do agree that this part of the model has potential for further development – in reality, the drag acting on particles depends on their shape, but also orientation, which will be more chaotic in

highly turbulent conditions, etc. We believe that an attempt to include those effects wouldn't change the main message of our present paper.

2. Sensitivity of results: The authors choose a particular OML-average volume fraction of 0.00168 for each category, so in total 0.005 (i.e. 0.5%). This is actually rather high. I think there should be better discussion of the sensitivity of results to this choice (e.g. Fmust increase with increasing ice concentration, but is the sensitivity linear or are there nonlinear feedbacks?)

When we planned our work, we decided that an analysis of how the OML mean ice volume fraction influences the modeling results will be left for the simulations with thermodynamics – they are perfect for studying how e.g. three-dimensional details of the OML circulation or the properties of frazil streaks change from an initial state without ice as the amount of ice gradually increases.
But we do agree that this point should be discussed in the present paper, and we add that issue to the discussion of the results.

3. Comparison with observations: The paper makes some comparison with observations, particularly the streaks of ice visible at the surface. However, the comparison is mostly qualitative. This is fairly well discussed in the final section; a forward link could be added in the final paragraph of page 20. A more quantitative comparison would be preferable. A starting point would be to devise and calculate statistical measures of the band size and spacing in the numerical calculations and then consider whether these are affected, for example, by wind speed. This could additionally be used to compare plots in Supplementary Fig. 9 quantitatively.

A more quantitative comparison to observations is possible when… observations are available! To the best of our knowledge, there are no published studies in which the geometry (shape, width, spacing, etc.) of frazil streaks is analyzed, not to mention studies analyzing relationships between the properties of frazil streaks and forcing conditions (wind, waves, etc.). In the papers that we cite throughout our manuscript, only very basic streak characteristics are mentioned, like e.g. an approximate spacing between streaks visible on a single satellite image or the fact that their width increases with increasing distance from the shore.
Therefore, as already mentioned on the first page, we are planning a separate study in which (i) satellite images will be analyzed in combination with meteorological data etc. in order to create an observational dataset containing frazil streaks + forcing information; (ii) statistical measures will be proposed suitable for characterizing streak geometry in both observations and modelling results, which will enable future model validation.

Specific comments

4. P2, L20: 'does have influence' ! 'influences'.

Changed.

5. P3, L6: suggest adding review article Daly: Frazil ice dynamics, CRREL Monograph, 84, 46 pp., 1984.

We added reference to that monograph.

6. P4, L14: suggest expanding discussion of laboratory observations.

We extend that fragment a bit in the revised manuscript.

7. P5, L3: explain briefly why turbulent conditions are necessary.

We made it clear that turbulent conditions lead to freezing within the water column, as opposed to freezing at the surface during calm conditions, leading to the formation of nilas.

8. P5, L4: parenthetical remark a bit confusing, I would delete whole remark and instead change 'buoyancy' to 'convection' or 'buoyancy-driven convection'.

We changed it as suggested.

9. P6, L10–12: is this good for frazil, especially the bigger crystals?

It is widely used even for very light "particles", like e.g. oil droplets, which have rising velocities larger than those considered in our study. Thus, although we do not know of a study analyzing it explicitly for sea ice crystals, we assume that this assumption is justified.

Yes, it has been corrected.

11. Sec. 3.3.3: I think this section could have had more discussion of uncertainty. I would imagine that (18) is a more robust relationship than the others.

Yes, it is true. We add some comments on that to the revised manuscript.

12. P15, L9: where does the latitude come into the calculation? I assume only in Coriolis term but the role of rotation didn't seem to be discussed much.

The Coriolis term occurs in equation (1) and it is defined below the set of equations (1)–(4).
We added some new results to the Supplementary Material, illustrating how the latitude influences the orientation of the frazil streaks, and we mention those results in the revised discussion, together with the directions of further research.

13. P15, L12: vertical boundary conditions on frazil concentration (I saw some earlier discussion of boundary conditions for CROCO in general, but presumably these references don't say anything about frazil).

The boundary conditions are the same as for other scalars and were therefore not mentioned. In our simulations we assumed no flux of frazil at the surface, but it might be non-zero in general (e.g., falling snow as a source of crystals). We added a corresponding sentence to the section 3.3.2.

14. P17, L1–4: how/why were these chosen? If you turn on crystal growth in future, results will be extremely sensitive to supercooling.

Yes, not surprisingly our model runs with thermodynamics are extremely sensitive to the $T$ and $S$ profiles. However, with thermodynamics switched off, the actual values of $T$ and $S$ are in fact not very important.

Important is the temperature difference between the ocean and the atmosphere (which we varied by modifying the air temperature), as it defines the buoyancy flux at the surface.

15. P17, L16: I would make it clearer that the phrase 'this choice' is referring only to the uniform distribution, not to all the other choices.

We made this statement more precise.

16. Fig. 5: Quite busy but just about readable, consider removing intermediate Ua.

Well, in the case of some variables shown in the plots the difference between 5 and 30 m/s is very large and it is good to have the 15 m/s case in between. We decided to leave that figure as it is.

17. P20, L12: expand discussion of interaction with pycnocline

We added some more comments on that.

18. Fig. 7: I found the main plots confusing and think they need a clearer x-label and caption. Is this a horizontal average? Do the PDFs integrate to 1?

We added information to the caption that should make it clear. Yes, the pdfs integrate to 1.

19. Sec. 5.2: This section gives an impression that buoyancy and drag are similarly important, but the graphs suggest that the all-forcing result is very similar to buoyancy, which suggests buoyancy is much more important than drag.

We take care in the revised text that the message in the text and in the figure is clear and unambiguous.

20. Fig. 9: Thin lines very hard to see and distinguish. I would make all lines thicker and use line style to distinguish.

We tried several version of those plots, but some lines simply do overlap and there is no way of making them all visible. But this is actually the message of the plot – some effects shown in them are negligible.

21. P24. L1: Typo? (Fig. 11g)?

No, there is no panel g in Fig.11, there are only a–d.

22. P24. L5: Typo in word 'important'.

Corrected.

23. P28. L19: In a different way to $F_\rho$?

In a different way in the sense that different aspects of the flow are affected. Taken alone, lower density due to higher ice concentration might lead to very turbulent rising of frazil –laden plumes, and increased viscosity might partially counteract that effect. But it is a pure speculation at this stage.

24. P30. L17: Could also mention flocculation?

We added flocculation to the list.

25. P30. L31: The editor may wish to consider the journal's policy about code availability. My opinion is that code by correspondence is less good (in terms of reproducibility) than code made publicly available with a doi.

We definitely agree!
In this case, it is planned that the frazil "toolbox" will be included in the publically available version of CROCO at some point in the future. Therefore, we decided not to place our present version of the code in any public repository, especially that it does not yet include informative comments and some parts are simply rough and ugly and implemented in a way that is not optimal computationally. And the thermodynamic part still requires additional testing.
But we are ready to share the code with anyone interested.

26. Supp. eq. (2): r appears on both LHS and RHS.

It is $r(e)$ on the l.h.s. and $r(e_s)$ on the r.h.s.
We wrote explicitly $r(e)$ in the revised version.

**Response to the comments of Reviewer #2 (H. Heorton)**

Can the authors comment on the chosen initial distribution of frazil ice crystals? I was expecting to see a sensitivity study of this chosen distribution and quantity of suspended crystals. I understand that this study is focussing on the interactions between crystals and the hydrodynamics with no thermodynamics implemented. Did the authors test other initial conditions? I think the paper needs a statement/discussion on the validity of the chosen distribution to allow the reader to understand the context of the simulations and to allow for the interpretation of these results amongst observations of frazil crystals within polynyas and other model simulations. For example in our paper Heorton (2017) we have many examples of the crystal concentrations at depth, although this is a 1D thermodynamic process model.

Many of the issues raised in this comment have been discussed in the first part of this document and we won't repeat them here.
In order not to leave those (fully justified) objections unanswered, we added some new results to the Supplementary Material, obtained with different proportions between the total mass of small and large crystals and with different total mass of ice. As already said, in our simulations with thermodynamics, on which we are working at present, there are many very interesting effects associated with the crystal size distribution, with both dynamic and thermodynamic processes involved, and we will investigate them in detail in a subsequent paper. With thermodynamics switched off, the effects are rather subtle and the main conclusions from our study remain unchanged.

Can the authors confirm whether they focus on latent heat or wind driven polynyas? The temperature conditions suggest a latent heat polynya where the temperature of the ocean restricts the growth of ice. But for these conditions what is the expected frazil crystal desnsity and how physically realistic are the choice of crystal concentration and ocean temperature? I realise that

these quantities are of secondary importance when thermodynamics are ignore and you focus on the crystal/flud interactions. However an idea of context will help future work where others may wish to compare your results to observations or other models.

The simulations include a range of situations with different wind/wave forcing and different ocean-atmosphere heat and moisture flux. In those simulations, in which the air temperature is very close to the water temperature, the dynamics is wind/wave driven, and the opposite is the case with weak wind and low air temperature. So it cannot be said that we are analyzing just one type of polynya. And it must be remembered that, without ice thermodynamics, the temperature and salinity are important only through their influence on water density and thus the vertical stability of the water column. In short, the buoyancy flux at the surface is the only relevant quantity influencing what is going on in the model, and it doesn't matter if it is associated with heat flux, moisture flux, or both.

I am also interested in the limitations of the model in terms of the intial conditions and time span. I will give some context to my question: We found in our paper that steady state condtions were possible for a wind opened polynya when modelling the thermodynamics properties of ice crystal growth (I am aware that you model a latent heat polynya). A crucial part of the steady state was secondary nucleation of small crystasls. This is crucial as small crystals have lower 'floatability' (as the authors here call it, a useful term) so can be more easily mixed downward, and thus replace the larger crystals that reached the surface (and were removed from the simulation in our model). This steady state was able to persist for several days of model simulation for particular model parameters and atmospheric forcing. However when chosing other parameters (see the results table in Heorton (2017), such a state was not achievable in the model. For example when reducing the OML turbulent mixing rates it was possible to precipitate all the frazil crytals to the surface and thus break the model. Balancing such phenomena was a major challenge of our model development and therefore I ask whether similar limitation were present in the authors model which contains very sophisticated modelling of the ocean turbulence and small scale flows. In particular, are there cases where all crystals collected at the surface? which model run has the greatest time variation in the vertical distribution of ice crystals? What are the model limitations due to the lack of thermodynamics? As the OML supercooling and secondary nucleation driven frazil crytal 'blooms' occur on time scales shorter than the model simulations shown here, how to the authors think that such processes will integrate with their model. I considered asking you to include time series plots alongside figures (5,6,9,10) to show how much variation occured for each simulation, though I don't think this is neccessary as the rest of the paper is so well presented and very extensive. However I think it a good idea to indicate how much variation from the initial conditions there are, for example how long did the surface increase in c3 for Ua=30 m/s Ta = -1.5 deg case in figure 6 (g,h,i) take to form and how stable was the feature? Did c3 continue to rise throughout the model run (with the layer of increased c3 getting increasingly shallow with time) or is the feature in the figures semi-stable? Similarly with figures (7,8,12), at which time point did such features start to occur and did they then remain for the rest of the model run? When running our model of frazil crystals in the OML, two runs with differing spatial distributions of crystal concentration, typically had differing patterns of time evolution. Do the authors find the same with their model?

Answering in detail all issues mentioned in this paragraph would require several pages, but we will try to discuss/answer the most important points.
We obtain quasi-stationary conditions in all our simulations (quasi-stationary meaning stable, time-independent area-averaged profiles of flow velocity, frazil volume fractions and other quantities

characterizing the bulk flow, but with constantly evolving individual Langmuir cells, convergence zones etc.).

And yes, there are cases – those with no or very weak wind and no or very small air-water temperature differences – in which the frazil accumulates within the uppermost layer of the model. This is just another case of a stable vertical profile. We do not quite understand in what sense that situation "breaks the model" in the simulations done by the reviewer. Is it a numerical instability? Otherwise, calm conditions with all frazil crystals rising with their terminal velocities until they reach the surface is realistic and leads to a stable state with non-zero frazil volume fraction at the top and zero everywhere. In arguably much more interesting cases when mixing is present, our model reaches quasi-stationary state as well, with less trivial vertical profiles of frazil – as those shown in the figures of our paper.

Reading your comments above, as well as those in the "minor points" section ("I doubt there are any equilibrium profiles of ice crystals") made us realize that we didn't pay enough attention to the existence of stable profiles in our paper. Although this might seem surprising, the reason is that this aspect of the results seemed quite obvious to us. Stable profiles of material transported within the OML have been observed and modelled for different substances, including very light ones, like e.g. oil droplets (with density of ca. $0.85 kg/m^3$). They also occur within the bottom boundary layer (sediments) and ABL (dust particles), and in all those settings result from a balance between particles' rising/sinking velocity and turbulent mixing. In our case, the individual clouds of frazil, associated with surface convergence zones, evolve in time, but the domain-averaged profiles are very stable. Considering high vertical velocities within downwelling zones (~0.1m/s are not exceptional, locally the values might be even higher), it is not surprising that the ice doesn't accumulate at the surface and that the small crystals are mixed throughout the OML.

Also, the model reaches the quasi-stationary state relatively fast. For example, if we assume vertical velocity of 0.1m/s, and the OML thickness of 100m (as in our simulations) the travel time from the bottom to the top of the OML is less than 20 minutes. Our observations suggest that the time necessary for the model to reach the quasi-stationary state is proportional to that time scale: after a few up-/downwelling cycles the bulk model state is close to stationary.

As you suggest, we do not add figures showing the time evolution of the model to the main text, but we decided to add them to the Supplementary Material.

Minor points
Pg1
L22 The use of 'obviously' is not very helpful in section with no references.

Some statements are so general that they are rather 'obvious' even without references, and in our opinion this is an example of such a statement, but we removed the word 'obviously'.

Pg2
L1-2 I agree with this sentence but it needs references for the situations listed. For example - Air-Ice-Ocean Interaction McPhee 2006

We added reference to McPhee 2008.

L3-5 again this sentence is good, but I'd like to see some references.

But this sentence describes what we are going to study in our work, it does not refer to any earlier results.

L5 'distinctive feature' definitely needs a reference

Yes. We added reference to the review paper on polynyas by Morales Maqueda et al (2004).

L11 and L 16 I think here you refer to sea-ice-climate models, or the sea ice component of a climate model

Not necessarily. We refer to continuum sea ice models that might be run on large- or regional-scale for climate studies or short-term predictions. In line 11, we think it is clear from the previous sentence, in which we speak of "continuum numerical sea ice models". In line 16, we added the word "continuum" so that there is no doubt what type of models is meant.

L 17 a reference to these observations is needed here where they are introduced
L 20 again a reference for the additional physical processes that are caused by the presence of frazil ice are needed.

This part of the text is an introduction to the following paragraphs, in which we describe the available observations and cite the relevant literature. We think that there is no need to list the same references here.

L 28 I see you have the references included here. A link from the above paragraph to the area of the paper with more detail is needed to aid a reader seeking greater detail.

But you speak about two paragraphs placed together in the text, one directly after the other – providing a link from the first one to the next one seems superfluous. The reader does not have to "seek" for those references in another part of the paper.

Pg4
L 19 - 21 consider splitting this sentence up as it is difficult to read. Also this sentence describes the limit of the presented modelling study. Crystal thermodynamics are not implemented. This needs to be very clearly stated.

We state the fact that thermodynamics is switched off at several locations in the text, not only in the introduction, but also in sections related to model description, setup and results, so that we think that no one reading the paper can have doubts regarding that issue.

Pg5
L5 'an earlier idea by' to be removed.

It has been removed.

L 12 can you add a definition for the Langmuir stability length

We added a short explanation to the text.

L 26 whilst you have a description of Langmuir turbulence above, a definition of Langmuir circulation and transport will be helpful here.
L 30 I see here you have linked the Langmuir circulation and turbulence. As this section contains many different terms, It would aid the reader to have definition and physics behind Langmuir circulation/turbulence/transport defined at the beginning.

Langmuir circulation and Langmuir turbulence actually mean the same thing. The first term has been used for a long time, the second one has been introduced more recently, as it has become clear that the OML circulation does not form those perfect, regular cells described in the early papers on the subject (that are visualized by many when they hear about "Langmuir circulation"), but instead constantly evolves in time, with individual cells forming, disappearing or joining with their neighbors. But Langmuir circulation as a general term is still widely used.

Pg 8
L 18 Does this description of low wind conditions over polynyas come from an observation or your results? Can you describe or give a reference.

Those sentences actually state very basic physics: if there is no advection of cold air, the lower ABL will get warmer in time due to large heat flux from the underlying ocean, gradually reducing the air-water temperature difference.

Pg 9
L7 Can you expand on 'preferential concentration of frazil within the turbulent OML should be limited to the largest crystals ' please? Do you mean that due to the increased rising velocity of larger ice crystals one only needs consider the largest crystals? If so I find this alarming as previous results show that a there are vastly greater numbers of small crystals for certain cases, particulalry at depth.

But this is exactly what we say: small crystals do not undergo preferential concentration, meaning that they are well mixed throughout the OML and thus can be found not only at the surface, but at depth as well. Why the fact that only the largest crystals tend to accumulate at the surface should mean that "one needs consider" only those crystals? The question of the reviewer suggests that we state something about the importance, or relevance of different size classes. We're just discussing where in the water column they can be found.
We slightly reformulated the controversial sentence and hope its meaning is clear now.

Equation 1-4, as the model description is complex, I would like you to briefly describe what physical quantity is being conserved in each of the governing equations. This will greatly aid the readability of the following sections.

We added that information before equations (1)-(4).

Pg 11
L 25, which equation describes the horizontal mixing?

There is no separate equation for horizontal mixing, as usual in hydrodynamic models it is a source term in the momentum equations.

Equation 12 in which equations are these transfer coefficients implemented?

As stated in the text: the full set of equations for the ocean-atmosphere fluxes can be found in the Supplement. The number of equations is large and details are not important for the present analysis – therefore they are not included in the main text.

L5 can you provide a reference for the Schiller Naumann model and use for frazil crystals.

The Schiller-Naumann model is very old (it was published in 1935) and it has long become a standard (a default option) in multi-phase modeling and related methods – so much so that the original paper is very rarely cited. We added a reference to a textbook by Kolev (2011).

L 6 what observations?

Those described in the following sentence, where we provide observed values from Guest and Davidson (1991).

Figure 4 In my experience frazil volume fractions rarely exceed 0.25, this is what is expected for grease ice, and within the OML will be lower. What is the application of the parameterisation in this figure where a volume fraction of 0 to 0.5 is presented?

Values as high as 0.44 were observed for example by Kaufmann & Martin (1981), and considered theoretically by deCarolis et al (2005). We agree that the typically found values are lower, but we see no reason to artificially put an upper limit of, say, 0.25 on the range of volume fractions considered here. We write in the text that the range <0.2 is relevant in most cases.

L 13 I see the choice of frazil crystal radius is sensible when considering the terminal velocities. However how does the choice of crystal classes and initial concentrations compare to previous studies? A resultant ice thickness of all suspended ice crystals of 0.5 m seems to be a large volume of ice to be suspended within the OML. Our work with frazil crystal modelling with thermodynamics suggests values or around 0.1 m (Heorton 2017). Also with climate models the initial stages of ice growth typically result in 0.05 m of new ice. Is your chosen crystal distribution specifically for a latent heat polynya? I realise that your model contains no thermodynamics so you need to seed a high amount of crystals.

In your model, the ice reaching the surface is removed from the model, isn't it? In our case, the total ice volume (or mass) includes ice suspended in the water column and ice staying at the surface. We agree that the amount of ice in our simulations is rather high, but it is not unrealistic.
As far as the last sentence is concerned: no, we don't have to "seed" any particular amount of crystals, the model can be run with extremely low ice concentrations without any problems. But the feedbacks between frazil and hydrodynamics are negligible if the amount of ice is very small.
As we already said earlier, the question of how the total amount of ice influences the OML state and circulation is very interesting and important, but we decided to analyze it based on the results of

simulations with thermodynamics – they are perfect for studying how the situation changes during ice growth.

Pg 18
Table 2 Are all parameters other than Ua and Ta derived from these choices and other model parameters? If so can you say so in the caption.

Yes, they are.
We added a note saying that the quantities in the table are computed according to the formulae in Supplementary Note S1.

Table 3 Can you expand on the what the F symbols on the left of the table indicate within the caption? If you're including this table to inform the reader about the what was included in the runs, please make it easy to read!

The table was meant as a quick reference for the reader, accessible more easily than the same information provided in the text.
We made the caption more informative.

L 7 is the wind aligned with the x or y axis?

Along the x-axis. We added the information on wind (and wave) direction to section 4.

Pg 20
L5 I doubt there are any equilibrium profiles of ice crystals. Can you comment on this, what are the time variations in the mean contours? Are the variation plots spatial or time variation? For your model with no thermodynamics I expect all the frazil ice to eventually be driven by buoyancy and collect at the surface.
L7 are you plots in figure 6 time averaged over 18 hours? Over this time scale I would expect a large change in vertical concentration. However my experience is from a thermodynamic model focusing on supercooling and ice crystal growth. Can you comment on the variance in time of the spatial coverage of ice crystals for the cases presented in figure 6?
L 31 as these patterns continually evolve in time, do the mean (over x and y directions) vertical profiles of crystal concentration and ocean velocity also vary in time? how long do the spatial patterns take to form?

See our response to one of your previous comments.

Pg 21
Fig 7 what time point were the snap shots taken? How representative are these snap shots of the model run? How do they compare to initial model conditions? How long do they take to form?

In accordance with our previous comments: the details of the patterns do evolve, but their domain-averaged statistical properties remain the same.

Pg 22
Fig 8, similar comments to fig 7. These two figures are well presented and look great, but I find myself asking how representative are they of the whole model run. Additional information about the

temporal changes observed in the results will help.

As above.

Pg 28
L2 which studies have shown this?

We added a reference to the review paper by Chamecki et al. (2019), where those studies are summarized.

Pg 29
L9 is this statement referencable?

Honestly speaking, this sentence is based on discussions of one of us (A.H.) with different scientists during workshops and conferences. The lecture (and subsequent discussion) by Steve Ackley during the "Mathematics of Sea Ice Phenomena" programme in Cambridge in 2017 would make a very good reference, but of course it is not a publication.
We will try to find some published references and decide whether to keep that sentence in the final paper version or to modify/remove it.

L14 can you link back to the the figure or discussion where this is presented?

We added references to the relevant figures.

L 34 is your discussions of high and low ice concentrations here relating to the difference between ocean with suspended crystals and a grease ice layer, or are you referring to the total ice concentration within the OML?

Low ice concentration means *locally* C<<1, everywhere in the model domain. The equations of all those models are not formulated for mixtures (see the comment of reviewer #1 regarding the (1-C)-terms in the momentum equations).

---

## Editor Decision (ED1)

**1   Summary**

This paper investigates the dynamics of the ocean mixed layer (OML) in the presence of frazil and grease ice using Large Eddy Simulation (LES). It studies the effect of wind-driven, convective and Langmuir turbulence on frazil ice, and also the effect of frazil ice on OML dynamics.  The turbulent flow leads to segregation of the frazil ice and the formation of streaks of ice on the surface, which are qualitatively similar to field observations. The frazil ice can strongly influence OML dynamics, primarily through its effect on buoyancy. The study suggests several avenues for future research.

I think that the topic of the study is interesting and novel in several aspects.  Previous observations have only been interpreted in a qualitative fashion and previous models have been one-dimensional rather than the three-dimensional calculations presented here. The paper is very well written and the analysis performed is thorough with most of the limitations clearly explained.  There are a few relatively small weaknesses discussed below which the authors can use to revise their manuscript. However, overall, I think the paper is excellent and should be *accepted subject to minor revisions*.

**2   General comments**

1. **Model formulation:**  there are some limitations/assumptions of the model that should be discussed more clearly or considered in further or future calculations.

   The hydrodynamic equations (1–4) assume that the concentration of frazil is small. If this were relaxed, they would need terms like $(1-C)$, where $C$ is the total frazil concentration, in various places (see e.g. Jenkins and Bombosch, 1995).

   The frazil model doesn't consider crystal growth (which is a reasonable starting point and is well discussed).  However, I didn't understand why only three crystal sizes were used rather than a much better-resolved crystal size distribution? Presumably, this is not a very expensive part of the overall calculation? Was the sensitivity to the number of crystal size classes tested? It will certainly be essential to include many more when crystal growth and nucleation are considered (as mentioned some of the cited references).  Another subtle issue is the assumption that the crystals have a constant aspect ratio.  An alternative is to assume they have a constant thickness, which is arguably more reasonable from a crystal growth point-of-view. The crystals remain disk-shaped because it is energetically much easier to grow radially than in thickness.

   The results presented here are clearly very sensitive to the frazil terminal veloc-

ity (figure 3). I think the authors should consider comparing their calculations with laboratory data (e.g. of McFarlane et al. 2014). They should also consider crystal-shape effects (assuming eq. 16 wasn't designed for disk-shaped particles).

2. **Sensitivity of results:** The authors choose a particular OML-average volume fraction of 0.00168 for each category, so in total 0.005 (i.e. 0.5%). This is actually rather high. I think there should be better discussion of the sensitivity of results to this choice (e.g. $F_\rho$ must increase with increasing ice concentration, but is the sensitivity linear or are there nonlinear feedbacks?)

3. **Comparison with observations:** The paper makes some comparison with observations, particularly the streaks of ice visible at the surface. However, the comparison is mostly qualitative. This is fairly well discussed in the final section; a forward link could be added in the final paragraph of page 20.

   A more quantitative comparison would be preferable. A starting point would be to devise and calculate statistical measures of the band size and spacing in the numerical calculations and then consider whether these are affected, for example, by wind speed. This could additionally be used to compare plots in Supplementary Fig. 9 quantitatively.

**3  Specific comments and technical corrections**

4. **P2, L20:** 'does have influence' → 'influences'.

5. **P3, L6:** suggest adding review article Daly: Frazil ice dynamics, CRREL Monograph, 84, 46 pp., 1984.

6. **P4, L14:** suggest expanding discussion of laboratory observations.

7. **P5, L3:** explain briefly why turbulent conditions are necessary.

8. **P5, L4:** parenthetical remark a bit confusing, I would delete whole remark and instead change 'buoyancy' to 'convection' or 'buoyancy-driven convection'.

9. **P6, L10–12:** is this good for frazil, especially the bigger crystals?

10. **P12, L9:** should 'd' etc be italicized?

11. **Sec. 3.3.3:** I think this section could have had more discussion of uncertainty. I would imagine that (18) is a more robust relationship than the others.

12. **P15, L9:** where does the latitude come into the calculation? I assume only in Coriolis term but the role of rotation didn't seem to be discussed much.

13. **P15, L12:** vertical boundary conditions on frazil concentration (I saw some earlier discussion of boundary conditions for CROCO in general, but presumably these references don't say anything about frazil).

14. **P17, L1–4:** how/why were these chosen? If you turn on crystal growth in future, results will be extremely sensitive to supercooling.

15. **P17, L16:** I would make it clearer that the phrase 'this choice' is referring only to the uniform distribution, not to all the other choices.

16. **Fig. 5:** Quite busy but just about readable, consider removing intermediate $U_a$.

17. **P20, L12:** expand discussion of interaction with pycnocline

18. **Fig. 7:** I found the main plots confusing and think they need a clearer $x$-label and caption. Is this a horizontal average? Do the PDFs integrate to 1?

19. **Sec. 5.2:** This section gives an impression that buoyancy and drag are similarly important, but the graphs suggest that the all-forcing result is very similar to buoyancy, which suggests buoyancy is much more important than drag.

20. **Fig. 9:** Thin lines very hard to see and distinguish. I would make all lines thicker and use line style to distinguish.

21. **P24. L1:** Typo? (Fig. 11g)?

22. **P24. L5:** Typo in word 'important'.

23. **P28. L19:** In a different way to $F_\rho$?

24. **P30. L17:** Could also mention flocculation?

25. **P30. L31:** The editor may wish to consider the journal's policy about code availability. My opinion is that code by correspondence is less good (in terms of reproducibility) than code made publicly available with a doi.

26. **Supp. eq. (2):** $r$ appears on both LHS and RHS.
* * *
**High-resolution simulations of interactions between surface ocean dynamics and frazil ice**

Review H. Heorton

This paper documents the development and simulations from a 3D model of the Ocean Mixed Layer OML, within a polynya. The paper is very well presented and I particularly appreciated the extended description of the developments of OML models that allowed for this study to be undertaken.

The paper includes an extended section describing existing methods of analysing the mixing regimes for the OML, and then applying them to an OML with suspended particles. This is then expanded for latent heat polynyas, with the likelihood of each mixing regime discussed. The authors conclude that all three mixing regimes are likely in coastal polynyas but Langmuir turbulence is likely to dominate.

Then follows a full description the model that is thorough and easy to follow for such a complex model.

Two groups of simulations are presented. First a model with no frazil to hydrodynamic coupling under various atmospheric forcing. The mean states of these model runs are described showing the structure of the ocean currents and distribution of frazil crystals. The surface distribution of crystals and lateral currents are presented and contrasted with observations. Then a second group of simulations are presented that show the effect of adding in frazil-related processes. The results from these runs are compared with the first group to show the influence of each process.

The main finding presented are:

Adding the influence of frazil crystals to the net density of the ocean has the greatest influence on the vertical profiles of ice crystals.

The observed surface ice crystal collection is dominated by the largest size class of crystal. These observed surface features, whilst mainly driven by Langmuir circulation are also the result of multi-scale processes.

The results show that ice crystals or different sizes will be 'sorted' by their floatibility, with larger crystals at the surface and smaller crystals at depth. This true in previous work and is also true for this complex 3d modelling study.

I recommend the paper for publication with first a few minor corrections as listed below (mainly relating to referencing within the paper and adding citations). Also I have a few more questions that occurred to me when reviewing the paper that I'd like to see addressed.

Can the authors comment on the chosen initial distribution of frazil ice crystals? I was expecting to see a sensitivity study of this chosen distribution and quantity of suspended crystals. I understand that this study is focussing on the interactions between crystals and the hydrodynamics with no thermodynamics implemented. Did the authors test other initial conditions? I think the paper needs a statement/discussion on the validity of the chosen distribution to allow the reader to understand the context of the simulations and to allow for the interpretation of these results amongst observations of frazil crystals within polynyas and other model simulations. For example in our

paper Heorton (2017) we have many examples of the crystal concentrations at depth, although this is a 1D thermodynamic process model.

Can the authors confirm whether they focus on latent heat or wind driven polynyas? The temperature conditions suggest a latent heat polynya where the temperature of the ocean restricts the growth of ice. But for these conditions what is the expected frazil crystal desnsity and how physically realistic are the choice of crystal concentration and ocean temperature? I realise that these quantities are of secondary importance when thermodynamics are ignore and you focus on the crystal/flud interactions. However an idea of context will help future work where others may wish to compare your results to observations or other models.

I am also interested in the limitations of the model in terms of the intial conditions and time span. I will give some context to my question: We found in our paper that steady state condtions were possible for a wind opened polynya when modelling the thermodynamics properties of ice crystal growth (I am aware that you model a latent heat polynya). A crucial part of the steady state was secondary nucleation of small crystasls. This is crucial as small crystals have lower 'floatability' (as the authors here call it, a useful term) so can be more easily mixed downward, and thus replace the larger crystals that reached the surface (and were removed from the simulation in our model). This steady state was able to persist for several days of model simulation for particular model parameters and atmospheric forcing. However when chosing other parameters (see the results table in Heorton (2017), such a state was not achievable in the model. For example when reducing the OML turbulent mixing rates it was possible to precipitate all the frazil crytals to the surface and thus break the model. Balancing such phenomena was a major challenge of our model development and therefore I ask whether similar limitation were present in the authors model which contains very sophisticated modelling of the ocean turbulence and small scale flows. In particular, are there cases where all crystals collected at the surface? which model run has the greatest time variation in the vertical distribution of ice crystals? What are the model limitations due to the lack of thermodynamics? As the OML supercooling and secondary nucleation driven frazil crytal 'blooms' occur on time scales shorter than the model simulations shown here, how to the authors think that such processes will integrate with their model. I considered asking you to include time series plots alongside figures (5,6,9,10) to show how much variation occured for each simulation, though I don't think this is neccessary as the rest of the paper is so well presented and very extensive. However I think it a good idea to indicate how much variation from the initial conditions there are, for example how long did the surface increase in c3 for Ua=30 m/s Ta = -1.5 deg case in figure 6 (g,h,i) take to form and how stable was the feature?  Did c3 continue to rise throughout the model run (with the layer of increased c3 getting increasingly shallow with time) or is the feature in the figures semi-stable? Similarly with figures (7,8,12), at which time point did such features start to occur and did they then remain for the rest of the model run? When running our model of frazil crystals in the OML, two runs with differing spatial distributions of crystal concentration, typically had differing patterns of time evolution. Do the authors find the same with their model?

Minor points

Pg1
L22 The use of 'obviously' is not very helpful in section with no references.

Pg2
L1-2 I agree with this sentence but it needs references for the situations listed. For example - Air-Ice-Ocean Interaction McPhee 2006

L3-5 again this sentence is good, but I'd like to see some references.
L5 'distinctive feature' definitely needs a reference

L11 and L 16 I think here you refer to sea-ice-climate models, or the sea ice component of a climate model

L 17 a reference to these observations is needed here where they are introduced

L 20 again a reference for the additional physical processes that are caused by the presence of frazil ice are needed.

L 28 I see you have the references included here. A link from the above paragraph to the area of the paper with more detail is needed to aid a reader seeking greater detail.

Pg4

L 19 - 21 consider splitting this sentence up as it is difficult to read. Also this sentence describes the limit of the presented modelling study. Crystal thermodynamics are not implemented. This needs to be very clearly stated.

Pg5

L5 'an earlier idea by' to be removed.

L 12 can you add a definition for the Langmuir stability length

L 26 whilst you have a description of Langmuir turbulence above, a definition of Langmuir circulation and transport will be helpful here.

L 30 I see here you have linked the Langmuir circulation and turbulence. As this section contains many different terms, It would aid the reader to have definition and physics behind Langmuir circulation/turbulence/transport defined at the beginning.

Pg 8

L 18 Does this description of low wind conditions over polynyas come from an observation or your results? Can you describe or give a reference.

Pg 9

L7 Can you expand on 'preferential concentration of frazil within the turbulent OML should be limited to the largest crystals ' please? Do you mean that due to the increased rising velocity of larger ice crystals one only needs consider the largest crystals? If so I find this alarming as previous results show that a there are vastly greater numbers of small crystals for certain cases, particulalry at depth.

Equation 1-4, as the model description is complex, I would like you to briefly describe what physical quantity is being conserved in each of the governing equations. This will greatly aid the readability of the following sections.

Pg 11

L 25, which equation describes the horizontal mixing?

Pg12

Equation 12 in which equations are these transfer coefficients implemented?

Pg 13

L5 can you provide a reference for the Schiller Naumann model and use for frazil crystals.

Pg 14

L 6 what observations?

Pg 15

Figure 4 In my experience frazil volume fractions rarely exceed 0.25, this is what is expected for grease ice, and within the OML will be lower. What is the application of the parameterisation in this figure where a volume fraction of 0 to 0.5 is presented?

Pg 16

L 13 I see the choice of frazil crystal radius is sensible when considering the terminal velocities. However how does the choice of crystal classes and initial concentrations compare to previous studies? A resultant ice thickness of all suspended ice crystals of 0.5 m seems to be a large volume

of ice to be suspended within the OML. Our work with frazil crystal modelling with thermodynamics suggests values or around 0.1 m (Heorton 2017). Also with climate models the initial stages of ice growth typically result in 0.05 m of new ice. Is your chosen crystal distribution specifically for a latent heat polynya? I realise that your model contains no thermodynamics so you need to seed a high amount of crystals.
L 17 did you vary the frazil ice concentration at all?

Pg 18
Table 2 Are all parameters other than Ua and Ta derived from these choices and other model parameters? If so can you say so in the caption.

Table 3 Can you expand on the what the F symbols on the left of the table indicate within the caption? If you're including this table to inform the reader about the what was included in the runs, please make it easy to read!

L 7 is the wind aligned with the x or y axis?

Pg 20
L5 I doubt there are any equilibrium profiles of ice crystals. Can you comment on this, what are the time variations in the mean contours? Are the variation plots spatial or time variation? For your model with no thermodynamics I expect all the frazil ice to eventually be driven by buoyancy and collect at the surface.

L7 are you plots in figure 6 time averaged over 18 hours? Over this time scale I would expect a large change in vertical concentration. However my experience is from a thermodynamic model focusing on supercooling and ice crystal growth. Can you comment on the variance in time of the spatial coverage of ice crystals for the cases presented in figure 6?

L 31 as these patterns continually evolve in time, do the mean (over x and y directions) vertical profiles of crystal concentration and ocean velocity also vary in time? how long do the spatial patterns take to form?

Pg 21
Fig 7 what time point were the snap shots taken? How representative are these snap shots of the model run? How do they compare to initial model conditions? How long do they take to form?

Pg 22
Fig 8, similar comments to fig 7. These two figures are well presented and look great, but I find myself asking how representative are they of the whole model run. Additional information about the temporal changes observed in the results will help.

Pg 28
L2 which studies have shown this?

Pg 29
L9 is this statement referencable?
L14 can you link back to the the figure or discussion where this is presented?
L 34 is your discussions of high and low ice concentrations here relating to the difference between ocean with suspended crystals and a grease ice layer, or are you referring to the total ice concentration within the OML?